# TOKENFORMER: RETHINKING TRANSFORMER SCALING WITH TOKENIZED MODEL PARAMETERS

**Haiyang Wang**[1,3]\*, **Yue Fan**[1]\* **Muhammad Ferjad Naeem**[2], **Yongqin Xian**[2],
**Jan Eric Lenssen**[1], **Liwei Wang**[3], **Federico Tombari**[2], **Bernt Schiele**[1]
[1]Max Planck Institute for Informatics, SIC  [2]Google  [3]Peking University

## ABSTRACT

Transformers have become the predominant architecture in foundation models due to their excellent performance across various domains. However, the substantial cost of scaling these models remains a significant concern. This problem arises primarily from their dependence on a fixed number of parameters within linear projections. When architectural modifications (*e.g.*, channel dimensions) are introduced, the entire model typically requires retraining from scratch. As model sizes continue growing, this strategy results in increasingly high computational costs and becomes unsustainable. To overcome this problem, we introduce Tokenformer, a natively scalable architecture that leverages the attention mechanism not only for computations among input tokens but also for interactions between tokens and model parameters, thereby enhancing architectural flexibility. By treating model parameters as tokens, we replace all the linear projections in Transformers with our token-parameter attention layer, where input tokens act as queries and model parameters as keys and values. This reformulation allows for progressive and efficient scaling without necessitating retraining from scratch. Our model scales from 124M to 1.4B parameters by incrementally adding new key-value parameter pairs, achieving performance comparable to Transformers trained from scratch while greatly reducing training costs. Code and models are available at https://github.com/Haiyang-W/TokenFormer.git.

## 1 INTRODUCTION

Designing a powerful neural network architecture is a long-standing goal in machine learning. Recent developments in foundation models (FMs) have shown the potential of Transformers (Vaswani et al., 2017) as a universal computational architecture. Thanks to their flexibility and scalability, Transformers have achieved state-of-the-art performance across various domains, including natural language processing (NLP) (Radford et al., 2018; Alec et al., 2019; Brown et al., 2020), visual modeling (Dosovitskiy et al., 2021; Liu et al., 2021), vision-language (Liu et al., 2023; Wang et al., 2024), graph representation (Ying et al., 2021), and 3D vision (Wang et al., 2023a;b).

Transformers typically divide the computation required to process a single token into two distinct parts: interactions with other input tokens (*token-token* interaction) and computations involving the model's parameters (*token-parameter* interaction). The attention mechanism (Vaswani et al., 2017) facilitates token-token interactions, allowing modern general-purpose foundation models to encode multi-modal data into a unified token sequence and effectively capture complex dependencies among them (Liu et al., 2023; Zhu et al., 2023; Wang et al., 2023d). Conversely, token-parameter computations rely heavily on linear projections (Dunford & Schwartz, 1988), where input tokens are multiplied by a fixed set of parameters. This prescribed design limits scalability because increasing the model size requires altering core architectural components, often necessitating retraining the entire model from scratch. As models grow larger, this results in excessive resource consumption, making it increasingly impractical. In this paper, we introduce a novel architecture that enhances the flexibility of token-parameter interactions, allowing for incremental scaling of model parameters and effectively reusing previously trained models, thus significantly reducing the training burden.

---

\*Equal contribution.

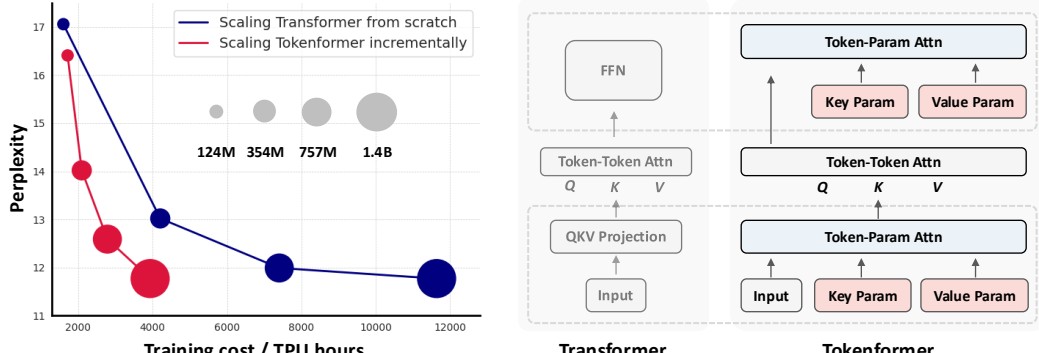

Figure 1: Traditionally, large transformer architectures are trained from scratch without reusing previous smaller-scale models (represented by blue dots on the left). In this paper, we propose a novel fully attention-based architecture that allows scaling model incrementally, thus greatly reducing the overall cost of training large transformer architectures (depicted by red dots on the left). The right panel delineates a comparison between conventional Transformer and our Tokenformer.

To achieve this objective, we introduce Tokenformer, a novel architecture that unifies the computations of token-token and token-parameter interactions by entirely employing the attention mechanism. The flexibility of our token-parameter attention layer, along with its ability to handle a variable number of parameters, inherently enhances the model's scalability, facilitating progressively efficient scaling.

As shown in Figure 1, we extend the Transformer architecture by preserving the computational patterns between input tokens while reformulating all the linear projections using a cross-attention mechanism. Specifically, to project features with input and output dimensions $D_1$ and $D_2$, we employ two sets of parameters, each comprising $N$ learnable tokens with channel dimensions of $D_1$ and $D_2$, respectively. In this formulation, input tokens serve as queries, and model parameters as keys and values. This flexibility renders our model's parameters inherently scalable with variable $N$, allowing for efficient expansion by continuously adding new key-value parameter pairs. Figure 1 shows that our model can be scaled incrementally from 124M to 1.4B parameters, achieving performance similar to training from scratch while saving more than half of the training cost.

The key contributions of this work are summarized as 1) As shown in Figure 1, we propose Tokenformer, a fully attention-driven neural network that treats model parameters as tokens, maximizing the flexibility of token-parameter computations while achieving competitive performance on standard benchmarks across both language and vision domains. 2) Thanks to this design, our model can be naturally scaled by progressively adding new key-value parameter pairs. Compared with the train-from-scratch approach (Biderman et al., 2023; Kaplan et al., 2020), our method achieves nearly the same performance while greatly reducing training costs.

We hope that the idea of tokenizing everything-whether it be data, parameter, or memory-and utilizing attention mechanisms to build connections between them will introduce a unique perspective on model architecture, inspiring innovative designs for future networks.

## 2 RELATED WORK

**Transformer** (Vaswani et al., 2017) has emerged as a foundational architecture in deep learning due to its versatile attention mechanism, enabling it to process any tokenized data and adapt to numerous domains, including language modeling (Radford et al., 2018; Touvron et al., 2023), image processing (Dosovitskiy et al., 2021), multi-modal understanding (Liu et al., 2023; Wang et al., 2024; 2023b; 2022), decision making (Chen et al., 2021b), graph learning (Yun et al., 2019), among others. While the Transformer effectively handles interactions among input tokens with flexibility, this property does not extend to computations involving model parameters, which are conducted via prescribed linear projections. In this work, we seek to restructure token-parameter interactions by developing a fully attention-based network that unifies both token-token and token-parameter computations through attention mechanisms, thus further extending the network's flexibility.

**Large Scale Training** has proven to be an effective approach for developing powerful foundation models. As demonstrated by models like the GPT series (Radford et al., 2018; Alec et al., 2019; Brown et al., 2020), simple architectures—when supported by larger training datasets and increased model sizes (measured in parameters)—often outperform more complex algorithms. Scaling up data is generally more cost-effective because it is independent of the model's architecture and allows for the continuous integration of new data through fine-tuning existing models (Kaplan et al., 2020). In contrast, increasing the model size often incurs extremely high costs, as it alters architectural details and usually requires retraining the entire dataset from scratch at each scaling step (Biderman et al., 2023). This significantly raises the expenses for building progressively larger models in the industry.

**Model Reusing.** Previous methods for reusing models have typically involved initializing larger models with pre-trained smaller models by duplicating (Chen et al., 2015; 2021a), stacking (Gong et al., 2019), or combining (Wang et al., 2023c) model weights. While these approaches can be effective, they often disturb the pre-established distribution of the smaller model, increasing the risk of losing pre-trained knowledge and slowing convergence. In contrast, our model allows for parameter scaling in a natural and seamless manner and preserves the integrity of the existing model.

## 3 METHODOLOGY

In this section, we first revisits the conventional attention mechanism in Section 3.1. Then, Section 3.2 introduces Tokenformer, a natively scalable architecture centered around a flexible token-parameter attention layer. Finally, incremental model scaling of Tokenformer is detailed in Section 3.3.

### 3.1 PRELIMINARIES

Transformer models (Vaswani et al., 2017) have established themselves as fundamental architectures in deep learning, demonstrating outstanding performance across a wide range of tasks. The cornerstone of their success is the self-attention mechanism, which allows the model to dynamically assess the importance of each token, efficiently modeling complex dependencies among them.

Given a set of $T$ input tokens $X \in \mathbb{R}^{T \times d}$ with channel dimension $d$, the self-attention block first derives input-dependent query $Q$, key $K$, and value $V$, with three distinct linear projections as

$$Q = X \cdot W^Q, \quad K = X \cdot W^K, \quad V = X \cdot W^V, \tag{1}$$

where the $W^Q, W^K \in \mathbb{R}^{d \times d_k}$ and $W^V \in \mathbb{R}^{d \times d_v}$ are learnable weight matrices. The attention scores are calculated by measuring the similarity between query and key vectors, followed by a softmax function to obtain normalized weights. These scores are subsequently used to compute the output of the scaled dot-product attention as,

$$\text{Attention}(Q, K, V) = \text{softmax}\left[\frac{Q \cdot K^\top}{\sqrt{d}}\right] \cdot V, \tag{2}$$

where $\sqrt{d}$ is a scale factor for alleviating small gradients caused by softmax. Finally, the output is,

$$O = X_{\text{att}} \cdot W^O, \tag{3}$$

with $X_{\text{att}}$ being the attention output and $W^O \in \mathbb{R}^{d_v \times d}$ as the output projection matrix.

The above architectural design enables the model to flexibly manage interactions between tokens of varying lengths, thereby allowing modern general models to concurrently process any form and quantity of tokenized multi-modal data. This capability markedly enhances the development of current AI domain and is fundamental to the success of transformer-based systems.

### 3.2 TOKENFORMER

Although transformers excel across various domains, their scalability is limited by high computational overheads resulting from prescribed token-parameter interactions (*i.e.*, linear projections). As a result, scaling strategies that adjust architectural components (*e.g.*, channel dimensions) typically require retraining the entire model from the beginning, leading to inefficient use of computational resources.

To overcome this challenge, we propose Tokenformer, an architecture entirely based on attention mechanisms. The central innovation of Tokenformer is token-**P**arameter **attention** (Pattention) layer,

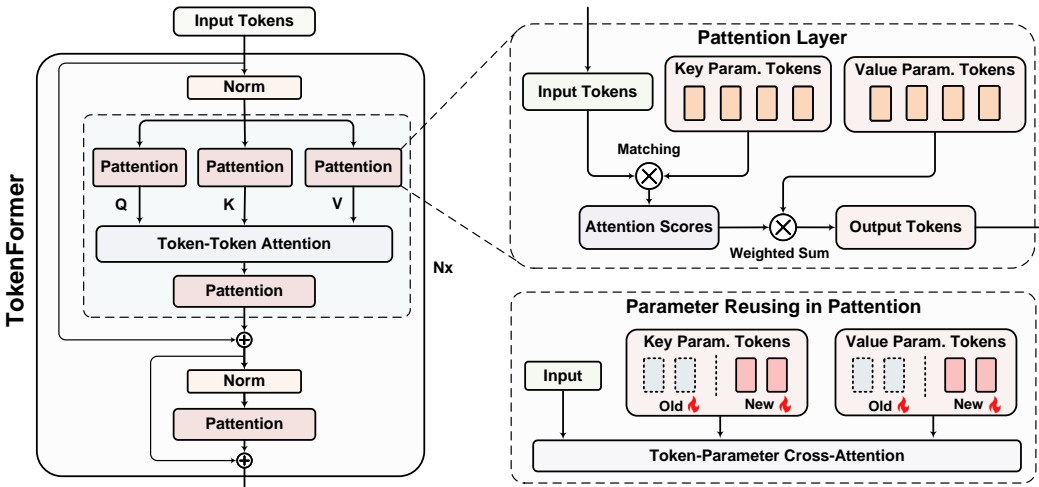

Figure 2: Tokenformer is a fully attention-driven architecture featuring a new token-**P**arameter **attention** (Pattention) layer. The Pattention uses a set of learnable tokens to represent model parameters and lets the input tokens attend to them. As the model scales, Tokenformer adds new learnable tokens to expand the existing key-value parameter sets, while keeping the feature dimension constant and leaving the rest of the computation unaffected.

which incorporates a set of trainable tokens functioning as model parameters and then employs cross-attention to manage interactions between input tokens and these parameter tokens. In this way, the Pattention layer introduces an additional dimension—the number of parameter tokens—which operates independently of the input and output channel dimensions. This decoupling enables input data to dynamically interact with a variable number of parameters, providing the flexibility required for incremental model scaling by reusing pre-trained models. Consequently, training larger models is greatly accelerated while achieving performance on par with transformers trained from scratch.

| Notation | Description | Notation | Description |
|---|---|---|---|
| $Q, K, V$ | Input-dependent query, key, value. | $(K_P^Q, V_P^Q), (K_P^K, V_P^K), (K_P^V, V_P^V)$ | Param tokens for generating $Q, K, V$. |
| softmax | Standard softmax: $\exp + L_1$norm | $\Theta$ | Modified softmax: $L_2$norm + gelu |

**Pattention Layer.** Let the input tokens and output tokens be represented as $\mathcal{I} \in \mathbb{R}^{T \times d_1}$ and $\mathcal{O} \in \mathbb{R}^{T \times d_2}$, where $T$ is the sequence length, and $d_1$ and $d_2$ are the input and output dimensions, respectively. To implement our Pattention mechanism, we introduce two sets of $n$ learnable parameter tokens: $K_P \in \mathbb{R}^{n \times d_1}$ representing the keys, and $V_P \in \mathbb{R}^{n \times d_2}$ representing the values. The output $\mathcal{O}$ from the scaled dot-product Pattention layer is computed as:

$$\text{Pattention}(X, K_P, V_P) = \Theta\left(X \cdot K_P^\top\right) \cdot V_P, \tag{4}$$

where $\Theta$ is a modified softmax operation for stable optimization of Pattention layer. Standard softmax involves two steps: applying an element-wise exponential to the input matrix and then performing L1 normalization along the token dimension. Our modified softmax replaces this with L2 normalization along the token dimension, followed by applying the GeLU activation to each element. This design improves gradient stability in our architecture, smooths the attention scores, and results in better performance compared to the standard softmax operation (see Appendix A and Table 4 for details).

Our Pattention layer employs a cross-attention mechanism to manage interactions between tokens and parameters, thereby fully preserving the adaptability characteristic of attention mechanisms. Similar to how self-attention in Transformer models handles sequences with variable lengths, our Pattention layer is designed to process a flexible number of parameters independently of the input and output channel dimensions used in feature projection. This allows network parameters to be expanded seamlessly along the parameter token axis, enabling the effective reuse of pre-trained weights and offering a naturally incremental manner for model scaling.

**Overall Architecture.** Figure 2 illustrates the architecture of Tokenformer. Given the input tokens $X_{\text{in}} \in \mathbb{R}^{T \times d}$, we follow the design of the pre-norm transformer, the computation for the output of a Tokenformer layer is represented as follows:

$$X_{\text{inter}} = X_{\text{in}} + \text{MHA}(\text{LN}(X_{\text{in}})), \tag{5}$$

$$X_{\text{out}} = X_{\text{inter}} + \text{FFN}(\text{LN}(X_{\text{inter}})), \tag{6}$$

where LN denotes the layer normalization (Ba, 2016; Zhang & Sennrich, 2019), and MHA and FFN refer to our modified multi-head self-attention and feed-forward layer, respectively.

In the multi-head self-attention block, for simplicity, we consider a single-head variant and set both $d_k$ and $d_v$ equal to $d$. Then we replace all the linear projections with our Pattention layers. Let $\text{LN}(X_{\text{in}})$ be denoted as $X$, this block is formulated as follows:

$$Q = \text{Pattention}(X, K_P^Q, V_P^Q), \;\; K = \text{Pattention}(X, K_P^K, V_P^K), \;\; V = \text{Pattention}(X, K_P^V, V_P^V), \tag{7}$$

$$X_{\text{att}} = \text{softmax}\left[\frac{Q \cdot K^\top}{\sqrt{d}}\right] \cdot V, \tag{8}$$

$$O_{\text{att}} = \text{Pattention}\left(X_{\text{att}}, K_P^O, V_P^O\right), \tag{9}$$

where Eq. 7 and 9 represent token-parameter attention while Eq. 8 represents token-token attention. The key-value parameter tokens for the QKV projections are $(K_P^Q, V_P^Q) \in \mathbb{R}^{n_q \times d}$, $(K_P^K, V_P^K) \in \mathbb{R}^{n_k \times d}$, $(K_P^V, V_P^V) \in \mathbb{R}^{n_v \times d}$, while $(K_P^O, V_P^O) \in \mathbb{R}^{n_o \times d}$ is used for the output projection layer.

For consistency and simplicity, the feed-forward block in Tokenformer utilizes a single Pattention Layer. Denote $\text{LN}(X_{\text{inter}})$ as $X_{\text{ffn}}$, and the FFN computation is given by:

$$O_{\text{ffn}} = \text{Pattention}\left(X_{\text{ffn}}, K_P^{\text{ffn}}, V_P^{\text{ffn}}\right), \tag{10}$$

where $(K_P^{\text{ffn}}, V_P^{\text{ffn}}) \in \mathbb{R}^{n_{\text{ffn}} \times d}$ are learnable key-value pairs for FFN block.

By designing the architecture in this manner, we represent all fundamental components-including both input data and model parameters—as tokens within the computational framework. This token-centric perspective allows the utilization of successful attention mechanisms to unify two primary computations within the transformer, token-token and token-parameter interactions, thereby establishing a fully attention-based neural network characterized by exceptional flexibility.

**Architecture Configurations.** Our model meticulously mirrors the hyper-parameter configuration of the standard Transformer architecture. Taking GPT-2 (Radford et al., 2018) as an exemplar, which features 12 Transformer layers and a hidden dimension of 768, our model replicates this configuration with identical layer counts and dimensionality. The number of key-value parameter pairs in both the query-key-value and output projections corresponds directly to the hidden dimension. In contrast, the FFN module utilizes four times the number of parameter pairs relative to the hidden size. This architectural alignment facilitates the initialization of our model's parameters using a pre-trained Transformer, thereby ensuring seamless integration into the Transformer pre-training ecosystem.

### 3.3 PROGRESSIVE MODEL SCALING

Our model demonstrates strong suitability for large-scale model training along the parameter axis, attributable to the versatile design of the Pattention layer, which allows for the incremental development of larger models by reusing parameters from smaller, pre-trained counterparts.

To facilitate understanding without compromising generality, we employ a single Pattention layer to exemplify the intricacies of model scaling. Consider an existing Tokenformer model equipped with a set of pre-trained key-value parameter tokens, denoted as $K_P^{\text{old}}, V_P^{\text{old}} \in \mathbb{R}^{n \times d}$. As shown in Figure 2, to scale the model, we augment this set by appending new key-value parameter tokens $K_P^{\text{new}}, V_P^{\text{new}} \in \mathbb{R}^{m \times d}$ as

$$K_P^{\text{scale}} = \left[K_P^{\text{old}}, K_P^{\text{new}}\right], \quad V_P^{\text{scale}} = \left[V_P^{\text{old}}, V_P^{\text{new}}\right], \tag{11}$$

where $[\cdot^{\text{old}}, \cdot^{\text{new}}]$ means the concatenation operation along the token dimension and $K_P^{\text{scale}}, V_P^{\text{scale}} \in \mathbb{R}^{(m+n) \times d}$ are scaled parameter sets. The forward pass of the scaled model is then defined as

$$O = \text{Pattention}\left(X, K_P^{\text{scale}}, V_P^{\text{scale}}\right). \tag{12}$$

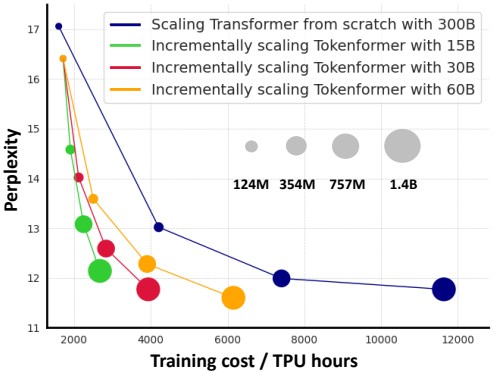 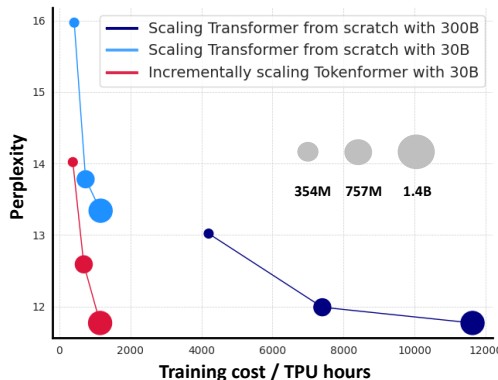

Figure 3: Evaluating model scaling costs through cumulative computational budgets. The Transformer baseline incurs expenses for each individual scaling step performed independently from scratch, whereas Tokenformer aggregates costs across all scaling stages, including training a 124M model initially, progressively scaling to 354M, 757M, and 1.4B parameters.

Figure 4: Evaluating model scaling costs by measuring the budget required at each scaling stage. The Transformer baselines used are consistent with those depicted in Figure 3, trained with 30B and 300B tokens. Similarly, for Tokenformer, the cost is the budget required for each incremental scaling step from a smaller one. All the experiments were conducted on TPU v4 hardware.

This scaling scheme permits the integration of an arbitrary number of parameters without altering the input or output dimensions. As demonstrated in Figure 3, this approach notably enhances training efficiency for models at greater scales without degrading performance. Importantly, if we choose to initialize $K_P^{\text{new}}$ with zero and $V_P^{\text{new}}$ with random values, similar to LoRA technique (Hu et al., 2022), our model can effectively resume from pre-training phase without losing the well-learned knowledge, facilitating faster convergence and accelerating the overall scaling process. Moreover, this manner provides a new dimension for scaling beyond just the hidden dimension, which helps maintain manageable token-token interaction costs and is well-suited for long-context modeling.

## 4 EXPERIMENTS

In this section, we present experimental results for the techniques described above. Section 4.1 validates the continual expansion capability of our model. Section 4.2 highlights the model's efficacy in handling tasks within both language and vision domains. Section 4.3 offers an in-depth comparison, highlighting our model's advantages over standard Transformer models. Finally, Section 4.4 details the ablation experiments conducted to assess the significance of each module in Tokenformer.

### 4.1 PROGRESSIVE MODEL SCALING

**Datasets.** Our models are trained using the OpenWebText Corpus described in (Gokaslan & Cohen, 2019). This corpus serves as a widely recognized open-source approximation of OpenAI's proprietary WebText dataset, which was employed in the development of GPT-2 (Alec et al., 2019). The dataset comprises textual data extracted from 8,013,769 Reddit-shared documents. During training, we randomly sample segments from these documents.

**Baseline Transformer Training from Scratch.** To evaluate the effectiveness of our progressive model scaling strategy, we established a baseline by training a Transformer model from scratch. Following the training procedures outlined in Karpathy (2022); Kaplan et al. (2020), we employed the AdamW optimizer (Loshchilov & Hutter, 2019) with a batch size of 512 sequences, each containing 1024 tokens. For a fair comparison with our incremental scaling approach, we configured two training variants based on the total number of training tokens. The first variant underwent $6 \times 10^5$ steps (approximately 300B tokens), consistent with the training steps utilized by Karpathy (2022) to replicate GPT-2 performance. The second variant was limited to $6 \times 10^4$ steps (approximately 30B tokens) to ensure comparability with each stage of our progressive scaling. In all trainings included

| Model | #Param | Pile ppl ↓ | LAMBADA ppl ↓ | LAMBADA acc ↑ | HellaSwag acc ↑ | PIQA acc ↑ | Arc-E acc ↑ | Arc-C acc ↑ | WinoGrande acc ↑ | Average acc ↑ |
|---|---|---|---|---|---|---|---|---|---|---|
| Pythia-160M (Biderman et al., 2023) | 160M | 29.64 | 37.25 | 35.4 | 30.3 | 62.3 | 43.6 | 23.6 | **51.3** | 40.1 |
| **Ours (TokenFormer-150M)** | 150M | **10.45** | **16.38** | **45.0** | **35.5** | **64.9** | **47.3** | **24.9** | 50.4 | **44.7** |
| Pythia-410M (Biderman et al., 2023) | 410M | 9.95 | 10.84 | 51.4 | 40.6 | 66.9 | 52.1 | 24.6 | 53.8 | 48.2 |
| **Ours (TokenFormer-450M)** | 450M | **8.28** | **7.69** | **57.3** | **47.5** | **69.5** | **56.2** | **26.7** | **54.6** | **52.0** |
| Pythia-1B (Biderman et al., 2023) | 1B | 7.82 | 7.92 | 56.1 | 47.2 | 70.7 | 57.0 | 27.1 | 53.5 | 51.9 |
| **Ours (TokenFormer-900M)** | 900M | **7.38** | **5.46** | **64.0** | **55.3** | **72.4** | **59.9** | **30.6** | **56.4** | **56.4** |
| GPT-Neo 1.3B (Black et al., 2021) | 1.3B | - | 7.50 | 57.2 | 48.9 | 71.1 | 56.2 | 25.9 | 54.9 | 52.4 |
| OPT-1.3B (Zhang et al., 2022) | 1.3B | - | 6.64 | 58.0 | 53.7 | 72.4 | 56.7 | 29.6 | 59.5 | 55.0 |
| Pythia-1.3B (Biderman et al., 2023) | 1.3B | 7.51 | 6.08 | 61.7 | 52.1 | 71.0 | 60.5 | 28.5 | 57.2 | 55.2 |
| **Ours (TokenFormer-1.5B)** | 1.5B | **6.91** | **5.24** | **64.7** | **60.0** | **74.8** | **64.8** | **32.0** | **59.7** | **59.3** |

Table 1: (**Zero-shot Evaluations.**) The best performance for each model size is highlighted in bold. Our comparisons are made with publicly available transformer-based LMs with various tokenizers. Following Pythia (Biderman et al., 2023), our model is trained for up to 300B tokens on pile dataset.

in our analysis, unless otherwise indicated, a learning rate of $6 \times 10^{-4}$ was employed, featuring a 2000-step warmup followed by a cosine decay to zero.

**Tokenformer with Progressive Model Scaling.** Building upon the above training protocols, we testify the performance of our model scaling with parameter sizes ranging from 124M to 1.4B. Unlike the aforementioned scratch-training approach, each scaling iteration leverages a pre-trained smaller Tokenformer to partially initialize the weights of the larger one described in Section 3.3. The scaling procedure begins with training the initial source model from scratch on approximately 300B tokens, mirroring the Transformer baseline. For scaling, we select the pre-trained model closest in parameter count to the target size for weight initialization. For example, to train a model with 354M parameters, we employ the 124M model as a partial initializer and retrain the entire model using a reduced computational budget (*e.g.*, 15B, 30B, or 60B tokens). This iterative process continues for scaling to 757M and then to 1.4B parameters. Notably, to simplify the scaling procedure, both new and existing parameters are trained equivalently with identical training hyperparameters throughout the process. Additionally, since the learning rate strategy in scaling follows a cosine schedule, the newly added parameters are randomly initialized to ensure a stable training process.

Our training optimizes the autoregressive log-likelihood (*i.e.*, cross-entropy loss) averaged over a 1024-token context and the log perplexity evaluated on the test set as the test score.

**Experimental Analysis.** As illustrated in Figure 3, our progressive scaling methodology employing Tokenformer achieves performance comparable to that of a Transformer model trained from scratch, while substantially reducing the training budget. Specifically, Starting with a 124M parameter model trained on 300B tokens, we progressively scaled to 354M, 757M, and 1.4B parameters, requiring only an additional 30B tokens—just one-tenth of the computational budget compared to the scratch-trained Transformer. This scaling process achieved a test perplexity of 11.77 at the 1.4B parameter level. In comparison, a Transformer model of the same size trained from scratch achieved a similar perplexity of 11.63 but with $3\times$ the training cost. Importantly, our approach reports cumulative training costs, encompassing all scaling stages, unlike the Transformer baseline that only accounts for individual stages. Even with this comparison, our method demonstrates a substantially lower computational cost than training a Transformer from scratch, thereby validating the effectiveness of our approach.

Figure 4 presents the training costs at each scaling stage for both our model and the standard Transformer. When compared to Figure 3, the cost savings are even more significant. Specifically, our model requires only one-tenth of the training costs associated with Transformer baselines. To mitigate the effects of varying training data, we also included the performance curve of a Transformer trained from scratch using an equivalent computational budget of 30B tokens. Under the same computational constraints, our progressively scaled model achieves a lower perplexity of 11.77 compared to the Transformer's 13.34, thereby highlighting the superior efficiency and scalability of our approach.

## 4.2 BENCHMARKING OF MODEL EXPRESSIVENESS

**Language Modeling.** We assess the efficacy of our proposed architecture through standard autoregressive language modeling tasks, benchmarking against existing transformer-based models. Evaluations are conducted using both pre-training metrics, specifically perplexity, and zero-shot performance measures. Training is performed on the Pile dataset (Gao et al., 2020), following the training protocol described in Biderman et al. (2023). Detailed training procedures and model sizes (depth and width) are provided in the Appendix F.

| Method | Image Size | #Param | Top-1 acc |
|---|---|---|---|
| ViT-B/16 (Dosovitskiy et al., 2021) | $384^2$ | 86M | 77.9 |
| DeiT-B/16 (Touvron et al., 2021) | $224^2$ | 86M | 81.8 |
| ViT-B/16 (MAE) (He et al., 2022) | $224^2$ | 86M | 82.3 |
| Ours-B/16$^\dagger$ | $224^2$ | 86M | 82.1 |
| **Ours-B/16** | $224^2$ | 109M | **82.5** |
| ViT-L/16 (Dosovitskiy et al., 2021) | $384^2$ | 307M | 76.5 |
| ViT-L/16 (MAE) (He et al., 2022) | $224^2$ | 307M | 82.6 |
| Ours-L/16$^\dagger$ | $224^2$ | 307M | 83.0 |
| **Ours-L/16** | $224^2$ | 407M | **83.1** |

Table 2: (**Image Classification.**) Comparison of standard vision transformer on ImageNet-1K. The training hyperparameters are completely consistent (batch size, learning rate, etc.) with He et al. (2022). $\dagger$ denotes models where the parameter size has been matched to that of the standard ViT.

| Operation | Parameter | | Training FLOPs | |
|---|---|---|---|---|
| | Transformer | Ours | Transformer | Ours |
| Embed | $n_{\text{vocab}}d_{\text{model}}$ | $n_{\text{vocab}}d_{\text{model}}$ | - | - |
| Attention: QKV Project | $3n_{\text{layer}}d_{\text{model}}^2$ | $2n_{\text{layer}}d_{\text{token}}(n_{\text{q}}+n_{\text{k}}+n_{\text{v}})$ | $6n_{\text{layer}}d_{\text{model}}^2 T$ | $4n_{\text{layer}}d_{\text{token}}(n_{\text{q}}+n_{\text{k}}+n_{\text{v}})T$ |
| Attention: Token-Token | | | $4n_{\text{layer}}d_{\text{model}}T^2$ | $4n_{\text{layer}}d_{\text{token}}T^2$ |
| Attention: Output Project | $n_{\text{layer}}d_{\text{model}}^2$ | $2n_{\text{layer}}d_{\text{token}}n_{\text{o}}$ | $2n_{\text{layer}}d_{\text{model}}^2 T$ | $4n_{\text{layer}}d_{\text{token}}n_{\text{o}}T$ |
| Feedforward | $8n_{\text{layer}}d_{\text{model}}^2$ | $2n_{\text{layer}}d_{\text{token}}n_{\text{ff}}$ | $16n_{\text{layer}}d_{\text{model}}^2 T$ | $4n_{\text{layer}}d_{\text{token}}n_{\text{ff}}T$ |
| De-embed | - | - | $2n_{\text{vocab}}d_{\text{model}}$ | $2n_{\text{vocab}}d_{\text{model}}$ |
| Total (Non-Embedding) | $N = 12n_{\text{layer}}d_{\text{model}}^2$ | $N = 2n_{\text{layer}}d_{\text{token}}(n_{\text{q}}+n_{\text{k}}+n_{\text{v}}+n_{\text{o}}+n_{\text{ff}})$ | $2NT + 4n_{\text{layer}}d_{\text{model}}T^2$ | $2NT + 4n_{\text{layer}}d_{\text{token}}T^2$ |

Table 3: Parameter counts and training compute estimates for Transformer and our Tokenformer. Sub-leading terms such as nonlinearities, biases, and layer normalization are omitted.

Table 1 presents the performance of Tokenformer across various widely-recognized zero-shot downstream tasks. Comparisons are drawn against leading open-source transformer models of equivalent scale, notably Pythia (Biderman et al., 2023), which utilizes the same tokenizer, dataset, and training duration (300B tokens) as our models. As shown in this table, our model achieves competitive performance compared to the standard Transformer, demonstrating the potential of our architecture in terms of expressive power as a foundation model.

**Visual Modeling.** Table 2 validates the expressiveness of our model in visual tasks. We compare our approach against the standard Vision Transformer (ViT) (Dosovitskiy et al., 2021) trained with supervised learning on the ImageNet-1K dataset (Deng et al., 2009). For a fair comparison, we used the MMDetection code base (MMDetection Contributors, 2018) and followed the hyperparameters and training strategy used in He et al. (2022). As shown in the table, our model achieves the same performance as ViT in visual modeling, confirming its expressiveness in visual tasks.

## 4.3 COMPARISON WITH STANDARD TRANSFORMER

Transformer can also achieve model reuse to a certain extent. Net2Net (Chen et al., 2015), a classical model growth method, proposes a technique to expand the width of neural networks by duplicating neurons. In this method, the pre-trained weight matrix of a transformer layer in the smaller model denoted $W_s^{\text{old}} \in \mathbb{R}^{d_s \times d_s}$, is used to create a larger weight matrix $W_l^{\text{new}} \in \mathbb{R}^{d_l \times d_l}$ $(d_l > d_s)$ to fill the larger model. This expansion is formulated as follows,

$$W_l^{\text{new}} = \begin{bmatrix} W_s^{\text{old}} & W_{l(12)}^{\text{new}} \\ W_{l(21)}^{\text{new}} & W_{l(22)}^{\text{new}} \end{bmatrix}, \tag{13}$$

where $W_{l(12)}^{\text{new}} \in \mathbb{R}^{(d_l-d_s) \times d_s}$, $W_{l(21)}^{\text{new}} \in \mathbb{R}^{d_s \times (d_l-d_s)}$, and $W_{l(22)}^{\text{new}} \in \mathbb{R}^{(d_l-d_s) \times (d_l-d_s)}$ are new parameters for expansion. The scaling procedures are the same as schemes introduced in Section 4.1.

**Controllable cost of token-token interaction for long-context modeling.** Recent advancements in Chain-of-Thought (CoT) modeling (Wei et al., 2022) have emphasized the critical importance of efficiently processing lengthy textual sequences (Tay et al., 2020) within Large Language Models (LLMs). As delineated in Section 1, the training costs of transformer architectures are primarily divided into two components: interactions involving model parameters and interactions among input sequences. Table 3 demonstrates that the computational complexity of transformer-based models exhibits a quadratic dependence on text length, scaling linearly with token-parameter interactions and quadratically with token-token interactions. Consequently, it is imperative to expand model parameters while controlling the computational burden of token-token interaction part.

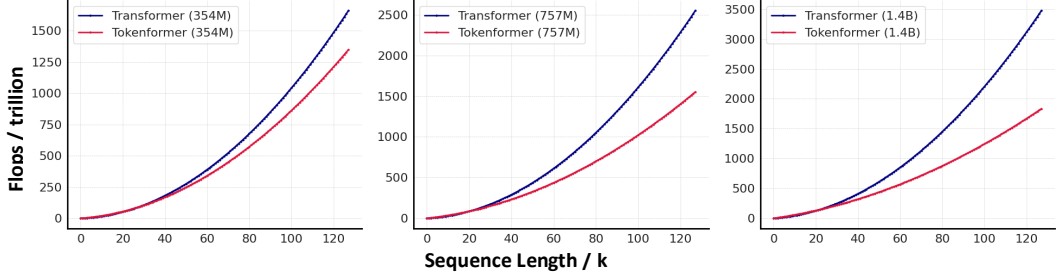

Figure 5: The relationship between FLOPs and text length for both Transformer and Tokenformer. As shown in Table 3, Transformer exhibits an increase in computational cost for token-token interactions as $d_{\text{model}}$ scales upwards. Our Tokenformer model, however, offers a flexible parameter scaling mechanism that maintains $d_{\text{token}}$ at a constant value. This strategy results in controllable computational costs for token-token interactions and markedly enhances the efficiency of long-text modeling.

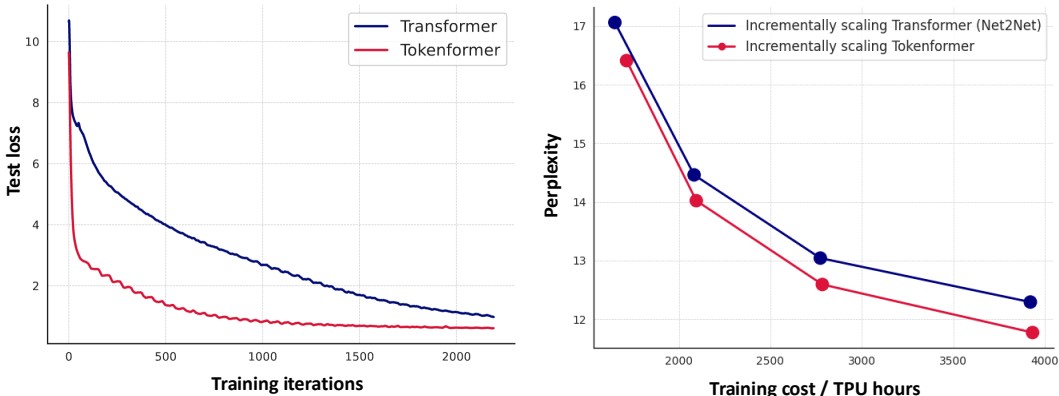

Figure 6: Loss curves comparing pre-trained Transformer and Tokenformer as their parameters are scaled during continued training on enwik8.

Figure 7: Performance benchmarking on incremental model scaling between Transformer with Net2Net scheme and our Tokenformer.

Conventionally, scaling transformer models involves increasing the channel dimension. For a fixed text length, this results in higher computational costs, mainly because dominant token-token interactions become more intensive, which hampers the model's performance with long texts. Our proposed model takes a different approach by decoupling the computation cost of token-token interactions from model scaling. We increase the parameter size without changing the token channel dimension, thereby maintaining the computational cost associated with token-token interactions. As shown in Figure 5, our model exhibits increasingly significant computational advantages over Transformers as the number of parameters grows, especially when processing longer sequences.

**Scaling without losing the well-learned distribution.** Our Tokenformer can maintain the existing output distribution when new key parameters are initialized to zero. This characteristic is beneficial for continuously scaling models to incorporate additional data, as it facilitates an increase in model capacity without disrupting the ongoing training process, thereby promoting rapid convergence.

To evaluate Tokenformer's scaling efficacy, we compare the loss curves of Net2Net-based transformer scaling against Tokenformer scaling. Both models, initially with 354M parameters, were pre-trained on the OpenWebText dataset. We then introduced the EnWik8 dataset and continued training with one epoch, expanding the models to 757M parameters to accommodate new data. Figure 6 demonstrates Tokenformer not only converges more rapidly but also reaches a lower final loss, attributable to its ability to preserve the output distribution during the resumption of training.

**Performance benchmarking on incremental scaling.** In this study, we progressively scale the standard Transformer using Net2Net approach detailed earlier. For a fair comparison, we aligned all hyperparameters, including the parameter size, learning rate, dataset, and so on. As shown in Figure 7, our model performs better in scaling compared to the standard Transformer.

## 4.4 ABLATION STUDY

| Nonlinear Function | Normalization | Top-1 acc |
|:---:|:---:|:---:|
| $e^x$ | $L_1$ Norm | 79.6 |
| GeLU | $L_1$ Norm | 81.7 |
| GeLU | $L_2$ Norm | 82.5 |

| Learnable Weight ($\gamma$) | Learnable Bias ($\beta$) | Top-1 acc |
|:---:|:---:|:---:|
| ✓ | ✓ | 82.6 |
| - | ✓ | 82.5 |
| - | - | 82.5 |

Table 4: Ablation of Softmax part on ImageNet classification with base model.

Table 5: Ablation of non-parametric layer normalization on ImageNet classification with base model.

**Optimized Softmax Function in Pattention Layer.** Within the token-parameter attention layer, we address training instabilities arising from the diminished gradients associated with the traditional softmax function. The conventional softmax operation comprises two primary steps: computing the exponential of attention scores followed by $L_1$ normalization. As shown in Table 4, to mitigate the issue of small gradients, we substitute the exponential non-linearity with the GeLU function, resulting in a performance enhancement of +2.1 points on the ImageNet classification benchmark. Subsequently, we replace the $L_1$ normalization with an $L_2$ normalization, yielding an additional improvement of +0.8 points, allowing our model to achieve performance parity with the standard ViT.

**Non-Parametric Layer Normalization.** In pursuit of enabling model expansion and the merging of two separately trained parameter token sets for subsequent studies, we modified the Transformer's layer normalization to a non-parametric variant by removing its trainable weights and biases. This adjustment guarantees that only the key-value parameters are subject to learning within the model. Empirical results presented in Table 5 demonstrate that the model maintains comparable performance after discarding the learnable weights and biases.

## 5 FUTURE WORK

**Extending the Mixture-of-Experts Paradigm.** We interpret Tokenformer as an extreme instantiation of the Mixture of Experts (MoE) framework (Fedus et al., 2022; Zhu et al., 2024), where each key-value parameter pair functions as an individual expert. This innovative MoE-like architecture has the potential to significantly reduce the computational costs associated with token-parameter interactions.

**Advancing Parameter-Efficient Tuning.** The scaling approach of Tokenformer, which involves integrating additional key-value parameter pairs, exemplifies a strategy for parameter-efficient tuning. When confronted with new tasks or datasets, the model can augment its pre-trained parameters by incorporating these new parameter tokens, thereby adapting to specific task requirements quickly.

**Integrating Vision and Language Models.** Leveraging the parameter-efficient tuning capabilities of Tokeformer, we can achieve seamless integration of visual and linguistic modalities. This can be accomplished by unifying the key-value parameter tokens derived from pre-trained visual Tokenformer and language Tokenformer into a single parameter set. Then, the new learnable tokens are introduced to perform vision-language alignment and instruction tuning.

**Enhancing Model Interpretability.** As Tokenformer is entirely based on attention mechanisms, it inherently benefits from the interpretability associated with attention in token-parameter interactions (Geva et al., 2021). This characteristic enhances the model's explainability, contributing to the AI community's efforts to develop more transparent and understandable models. Please see G.2 for more discussions.

## 6 CONCLUSION

This paper introduces Tokenformer, a naturally scalable architecture that leverages the attention mechanism to facilitate not only inter-token computations but also interactions between tokens and model parameters, thereby enhancing architectural flexibility. By representing model parameters as tokens, we replace all linear projections in Transformer with our Pattention, allowing for seamless incremental scaling without retraining from scratch. We believe that this architecture, offering greater flexibility than Transformers, will further contribute to the development of foundation models.

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

## A  GRADIENT OF PATTENTION LAYER

Our token-parameter attention mechanism employs $L_2$-normalization followed by the GeLU (Hendrycks & Gimpel, 2016) activation function, in contrast to the conventional SoftMax function in standard token-token attention layers, which utilizes an exponential transformation followed by $L_1$-normalization. This design choice is motivated by our experimental observation that SoftMax tends to increase the magnitude of outputs, often pushing them into regions where the gradients become extremely small, leading to an inferior overall performance (see Table 4).

Specifically, given a query token and $n$ key-value pairs with dimension $d$, let the similarity scores between the query and key tokens be represented as $A \in \mathbb{R}^{1 \times n}$. In standard SoftMax attention, the attention scores $S \in \mathbb{R}^{1 \times n}$ are computed as follows:

$$S_i = \frac{\exp(A_i/\sqrt{d})}{\sum_{j=1}^{n} \exp(A_j/\sqrt{d})}, \quad \forall\, i \in 1...n, \tag{14}$$

The derivative of the SoftMax function with respect to $A_i$ is given by:

$$\frac{\partial S_i}{\partial A_j} = \frac{1}{\sqrt{d}} S_i(\mathbb{1}_{i=j} - S_j) = \begin{cases} \dfrac{1}{\sqrt{d}} S_i(1 - S_j) & i = j \\[2mm] -\dfrac{1}{\sqrt{d}} S_i S_j & i \neq j. \end{cases} \tag{15}$$

In contrast, our activation function uses L2-normalization followed by the GeLU function. Denoting GeLU as $f$, the attention scores $Z \in \mathbb{R}^{1 \times n}$ are computed as:

$$\hat{S}_i = f(Z_i) = f\left(\frac{A_i \times \sqrt{n}}{\sqrt{\sum_{j=1}^{n} |A_j|^2}}\right), \quad \forall\, i \in 1...n, \tag{16}$$

For the derivative of our attention function when $i = j$, we have:

$$\frac{\partial \hat{S}_i}{\partial A_i} = f'\sqrt{n}\frac{||A||_2 - A_i\frac{A_i}{||A||_2}}{||A||_2^2} \tag{17}$$

$$= f'\sqrt{n}\frac{||A||_2 - A_i^2\frac{1}{||A||_2}}{||A||_2^2} \tag{18}$$

$$= f'\sqrt{n}\frac{1}{||A||_2}\frac{||A||_2^2 - A_i^2}{||A||_2^2} \tag{19}$$

$$= f'\sqrt{n}\frac{1}{||A||_2}\left(1 - \frac{Z_i^2}{n}\right) \tag{20}$$

$$= f'\frac{1}{\sqrt{n}}\frac{1}{||A||_2}(n - Z_i^2) \tag{21}$$

When $i \neq j$, the derivative becomes:

$$\frac{\partial \hat{S}_i}{\partial A_i} = -f'\sqrt{n}\frac{A_i\frac{A_j}{||A||_2}}{||A||_2^2} \tag{22}$$

$$= -f'\sqrt{n}\frac{1}{||A||_2}\frac{A_i A_j}{||A||_2^2} \tag{23}$$

$$= -f'\frac{1}{\sqrt{n}}\frac{1}{||A||_2}Z_i Z_j \tag{24}$$

Thus, the derivative of our attention function is:

$$\frac{\partial \hat{S}_i}{\partial a_j} = \begin{cases} f'\dfrac{1}{\sqrt{n}}\dfrac{1}{||A||_2}(n - Z_i Z_j) & i = j \\[3mm] -f'\dfrac{1}{\sqrt{n}}\dfrac{1}{||A||_2}Z_i Z_j & i \neq j. \end{cases} \tag{25}$$

Comparing the gradients of SoftMax (Eq. 15) and our method (Eq. 25), the key difference is that our gradient depends on the product $Z_i Z_j$, whereas SoftMax relies on $S_i S_j$. Due to the exponential nature of SoftMax, the distribution of $S_i$ tends to be sharper and more concentrated, which often drives the gradients toward zero. Conversely, our activation function produces a smoother distribution of $Z$, mitigating the vanishing gradient problem and enabling more stable training dynamics.

## B  ZERO INITIALIZATION IN TOKENFORMER

As shown in Section 3.3, during model scaling, initializing new key parameters to zero allows the model to continue training with minimal disruption. This is because zero initialization preserves the model's original output distribution, preventing significant interference to the learned representations.

In this section, we demonstrate that replacing the exponential function in the standard softmax with GeLU makes the Pattention layer invariant to newly added parameters when they are zero-initialized. This behavior cannot be achieved with the standard exponential-based softmax, as it is impossible to initialize new parameters in a way that guarantees the output scores of the newly added components are exactly zero after applying the exponential function, thereby preventing any influence on the existing parameter tokens.

Let $X \in \mathbb{R}^d$ be the input vector, and let the Pattention layer have $n$ key-value pairs, represented as $K_P, V_P \in \mathbb{R}^{n \times d}$. We fix the scale factor $\tau$ and, for simplicity of formulation, temporarily set it to 1. The output of the Pattention layer is computed as:

$$A = K_P \cdot X, \tag{26}$$

$$S = f\left(\frac{A}{\sqrt{\sum_j A_j^2}}\right), \tag{27}$$

$$O = V_P^\top \cdot S. \tag{28}$$

where $A$ is the score derived from $K_P \cdot X$ and $f$ is a non-linearity function, which in our formulation is set to the GeLU function (Hendrycks & Gimpel, 2016). When scaling the model by adding $m$ new key-value pairs with zero initialization, the output becomes:

$$\hat{A} = \begin{bmatrix} K_P \\ 0, .., 0 \\ \vdots \\ 0, .., 0 \end{bmatrix} \cdot X = \begin{bmatrix} A \\ 0 \\ \vdots \\ 0 \end{bmatrix}, \tag{29}$$

$$\hat{S} = f\left(\frac{\hat{A}}{\sqrt{\sum_j \hat{A}_j^2}}\right) = \begin{bmatrix} S \\ 0 \\ \vdots \\ 0 \end{bmatrix}, \tag{30}$$

$$\hat{O} = \left[V_P^\top, V_P^{\text{new}\top}\right] \cdot \hat{S} = O. \tag{31}$$

Since the newly added key parameters are initialized to zero, the attention mechanism does not modify the original output. Therefore, the output $\hat{O}$ remains identical to $O$. This property is advantageous for scaling models because it increases the model capacity without interrupting the well-learned distribution and the ongoing training process, leading to faster convergence.

## C  TABULAR MAIN RESULTS

Here, we provide the tabulated results corresponding to Figure 3 from the main paper. Table 7 presents the perplexity on the validation set of OpenWebText. The Transformer models are trained from scratch, while Tokenformer models leverage parameter reuse from smaller models (except for the first Tokenformer model with 124M parameters, which is also trained from scratch). We observe that Tokenformer achieves on-par performance to Transformers trained from scratch, but

| Perplexity | #Param | | |
|---|---|---|---|
| | 354M | 757M | 1.4B |
| Transformer Train from scratch 300B | 13.02 | 11.99 | 11.63 |
| Transformer Train from scratch 30B | 15.97 | 13.78 | 13.34 |
| Tokenformer Parameter Reusing 30B | **14.02** | **12.59** | **11.77** |

Table 6: Tabular results of Figure 4. The perplexity of models trained with different numbers of schemes is compared. Transformers are trained from scratch, while Tokenformer are progressively scaled up via parameter resuing. When trained with the same number of tokens (30B), Tokenformer demonstrates superior performance.

| #Param | Tokenformer | | Transformer | |
|---|---|---|---|---|
| | Tokens | Perplexity | Tokens | Perplexity |
| 124M | 60B | **16.41** | 300B | 17.06 |
| 354M | 15B | 14.58 | | |
| 354M | 30B | 14.02 | 300B | **13.02** |
| 354M | 60B | 13.59 | | |
| 757M | 15B | 13.08 | | |
| 757M | 30B | 12.59 | 300B | **11.93** |
| 757M | 60B | 12.28 | | |
| 1.4B | 15B | 12.14 | | |
| 1.4B | 30B | 11.77 | 300B | 11.63 |
| 1.4B | 60B | **11.60** | | |

Table 7: Tabular results of Figure 3. Due to parameter reusing, Tokenformer achieves the same performance while using a much lower number of training tokens when scaling up model sizes.

| method | model | 124M (60B) | | 354M (15B) | | 757M (15B) | |
|---|---|---|---|---|---|---|---|
| | | PPL | Flops | PPL | Flops | PPL | Flops |
| Net2Net | Transformer | 16.4 | 3.6 | 14.3 | 6.8 | 12.1 | 10.1 |
| HyperCloning | Transformer | 16.4 | 3.6 | 14.1 | 6.8 | 12.0 | 10.1 |
| Ours | Tokenformer | **16.1** | 3.6 | **13.7** | **3.6** | **11.5** | **3.6** |

Table 8: More comparison results of model scaling. The unit of Flops is $10^4 \times T^2$. Our method outperforms HyperCloning (Samragh et al., 2024) and Net2Net (Chen et al., 2015) while requiring less computational effort for token-token interactions.

with significantly reduced training costs due to parameter reuse. Notably, the largest Tokenformer (1.4B) achieves even a slightly better perplexity than its Transformer counterpart (11.60 vs. 11.63).

Table 6 compares Transformer models trained from scratch and Tokenformer trained by parameter reusing with varying amounts of seen tokens during training. It is evident that Transformers trained with the same number of seen tokens do not reach the performance level of Tokenformer with parameter reusing. This demonstrates that parameter reusing successfully transfers knowledge from smaller models, reducing training time without sacrificing performance.

# D    EXPERIMENTAL DETAILS ON PROGRESSIVE MODEL SCALING

In this experiment, we utilize the OpenWebText dataset (Gokaslan & Cohen, 2019) to evaluate the model scaling capability. The dataset comprises 8,013,769 documents, from which we randomly select 5% to serve as the validation set and report perplexity on this subset. We investigate four model sizes: 124M, 354M, 757M, and 1.4B parameters. Please find model specifications in Table 9. During parameter reusing of Tokenformer, we partially resume the old model parameters and add new key-value pairs to the Pattention layers and do not alter the number of layers or feature dimensions.

The training recipe is the same for both Transformers and Tokenformer: We do not implement dropout in either model and the logits are computed at the final layer using the embedding layer. The tokenizer is from GPT-NeoX-20B (Black et al., 2022). We employ the AdamW optimizer (Loshchilov, 2019) with $\beta_1 = 0.9$ and $\beta_2 = 0.95$. A learning rate of $6 \times 10^{-4}$ is employed, with a linear warmup over 2000 steps followed by a cosine decay to zero. The training is conducted with a batch size of 512 and a sequence length of 1024.

| Model | Layers | Hidden size | Attention KV Pairs | FFN KV Pairs | Heads | #Params |
|---|---|---|---|---|---|---|
| *Language Modeling* | | | | | | |
| Tokenformer-150M | 12 | 768 | 768 | 3072 | 12 | 150M |
| Tokenformer-450M | 24 | 1024 | 1024 | 4096 | 16 | 450M |
| Tokenformer-900M | 32 | 1280 | 1280 | 5120 | 16 | 900M |
| Tokenformer-1.5B | 40 | 1536 | 1536 | 6144 | 16 | 1.5B |
| *Visual Modeling* | | | | | | |
| Tokenformer-Base[†] | 12 | 768 | 576 | 2304 | 12 | 86M |
| Tokenformer-Base | 12 | 768 | 768 | 3072 | 12 | 109M |
| Tokenformer-Large[†] | 24 | 1024 | 768 | 768 | 16 | 307M |
| Tokenformer-Large | 24 | 1024 | 1024 | 4096 | 16 | 407M |
| *Parameter Reusing* | | | | | | |
| Tokenformer-124M | 12 | 768 | 576 | 2304 | 12 | 124M |
| Tokenformer-354M | 12 | 768 | 2140 | 8560 | 12 | 354M |
| Tokenformer-757M | 12 | 768 | 4850 | 19400 | 12 | 757M |
| Tokenformer-1.4B | 12 | 768 | 8620 | 34480 | 12 | 1.4B |

Table 9: Details of Tokenformer model variants used in Section 4. † indicates models whose key-value pairs are chosen to match the parameter numbers of Transformer of equivalent sizes.

# E    EXPERIMENTAL DETAILS ON SCALING WITHOUT LOSING THE WELL-LEARNED DISTRIBUTION

In this experiment, we utilize the EnWik8 (Mahoney, 2011) dataset to evaluate the model's capacity for continued adaptation to new data. The EnWik8 dataset comprises the first 100 million bytes of English Wikipedia in XML format.

We begin with a model size of 354M parameters, which has been pre-trained on OpenWebText (Gokaslan & Cohen, 2019), and then scale it to 757M parameters for both the Transformer and Tokenformer models. In the case of the Transformer, the 354M and 757M models differ solely in feature dimension and the number of heads in the multi-head attention mechanism, and we follow Net2Net(Chen et al., 2015) methodology for parameter expansion. For Tokenformer, we increase the number of key-value pairs from 2140 to 4850.

We employ a constant learning rate of $6 \times 10^{-4}$ and process the dataset for a single pass, resulting in a total of 2204 training steps. The AdamW optimizer (Loshchilov, 2019) is utilized, and we do not resume the optimizer's internal state from any previous runs. The batch size is set to 512, consistent with the batch size used during pre-training.

# F    EXPERIMENTS ON LANGUAGE MODELING BENCHMARKING

We evaluate the expressiveness and performance of our proposed architecture using standard autoregressive language modeling benchmarks, comparing its results to those of existing open-source LLMs, including RNN-based methods (Peng et al., 2023; Gu & Dao, 2023), in Table 10. Evaluations are conducted using both pre-training metrics, specifically perplexity, and zero-shot performance measures. For training, we used the Pile dataset (Gao et al., 2020) over a single epoch, adhering to the training strategy used by Biderman et al. (2023). The Adam optimizer is applied with a $6 \times 10^{-4}$ learning rate, and each batch contains 1024 samples with a sequence length of 2048 tokens, mirroring the setup of Mamba. The training process consists of 14,300 steps, with a 1430-step warmup, equivalent to 1% of total training steps. Detailed model specifications are in Table 9.

# G    DISCUSSIONS

## G.1    COMPARISON WITH A TWO-LAYER MLP

We aim to establish a novel and unified concept for model computation by treating all components, including parameters, as tokens. This token-based design allows the attention mechanism to perform dynamic interactions among them, such as data and parameter tokens, fostering greater flexibility and scalability in computational models.

| Model | #Param | Pile ppl ↓ | LAMBADA ppl ↓ | LAMBADA acc ↑ | HellaSwag acc ↑ | PIQA acc ↑ | Arc-E acc ↑ | Arc-C acc ↑ | WinoGrande acc ↑ | Average acc ↑ |
|---|---|---|---|---|---|---|---|---|---|---|
| Hybrid H3-130M (Fu et al., 2023) | 130M | - | 89.48 | 25.77 | 31.7 | 64.2 | 44.4 | 24.2 | 50.6 | 40.1 |
| Pythia-160M (Biderman et al., 2023) | 160M | 29.64 | 37.25 | 35.4 | 30.3 | 62.3 | 43.6 | 23.6 | 51.9 | 40.6 |
| Mamba-130M (Gu & Dao, 2023) | 130M | 10.56 | **16.07** | 44.3 | 35.3 | 64.5 | **48.0** | 24.3 | 51.9 | 44.7 |
| **Ours (TokenFormer-150M)** | 150M | **10.45** | 16.38 | **45.0** | **35.5** | **64.9** | 47.3 | **24.9** | 50.4 | **44.7** |
| Hybrid H3-360M (Fu et al., 2023) | 360M | - | 12.58 | 48.0 | 41.5 | 68.1 | 51.4 | 24.7 | 54.1 | 48.0 |
| Pythia-410M (Biderman et al., 2023) | 410M | 9.95 | 10.84 | 51.4 | 40.6 | 66.9 | 52.1 | 24.6 | 53.8 | 48.2 |
| Mamba-370M (Gu & Dao, 2023) | 370M | 8.28 | 8.14 | 55.6 | 46.5 | 69.5 | 55.1 | **28.0** | 55.3 | 50.0 |
| **Ours (TokenFormer-450M)** | 450M | **8.28** | **7.69** | **57.3** | **47.5** | **69.5** | **56.2** | 26.7 | 54.6 | **52.0** |
| Pythia-1B (Biderman et al., 2023) | 1B | 7.82 | 7.92 | 56.1 | 47.2 | 70.7 | 57.0 | 27.1 | 53.5 | 51.9 |
| Mamba-790M (Gu & Dao, 2023) | 790M | **7.33** | 6.02 | 62.7 | 55.1 | 72.1 | **61.2** | 29.5 | 56.1 | 56.1 |
| **Ours (TokenFormer-900M)** | 900M | 7.38 | **5.46** | **64.0** | **55.3** | **72.4** | 59.9 | **30.6** | **56.4** | **56.4** |
| GPT-Neo 1.3B (Black et al., 2021) | 1.3B | - | 7.50 | 57.2 | 48.9 | 71.1 | 56.2 | 25.9 | 54.9 | 52.4 |
| Hybrid H3-1.3B (Fu et al., 2023) | 1.3B | - | 11.25 | 49.6 | 52.6 | 71.3 | 59.2 | 28.1 | 56.9 | 53.0 |
| OPT-1.3B (Zhang et al., 2022) | 1.3B | - | 6.64 | 58.0 | 53.7 | 72.4 | 56.7 | 29.6 | 59.5 | 55.0 |
| Pythia-1.3B (Biderman et al., 2023) | 1.3B | 7.51 | 6.08 | 61.7 | 52.1 | 71.0 | 60.5 | 28.5 | 57.2 | 55.2 |
| RWKV-1.5B (Peng et al., 2023) | 1.5B | 7.70 | 7.04 | 56.4 | 52.5 | 72.4 | 60.5 | 29.4 | 54.6 | 54.3 |
| Mamba-1.4B (Gu & Dao, 2023) | 1.4B | **6.80** | 5.04 | **64.9** | 59.1 | 74.2 | 65.5 | 32.8 | **61.5** | **59.7** |
| GPT-Neo 2.7B (Black et al., 2021) | 2.7B | - | 5.63 | 62.2 | 55.8 | 71.1 | 61.1 | 30.2 | 57.6 | 56.5 |
| Hybrid H3-1.3B (Fu et al., 2023) | 1.3B | - | 11.25 | 49.6 | 52.6 | 71.3 | 59.2 | 28.1 | 61.4 | 58.0 |
| OPT-2.7B (Zhang et al., 2022) | 2.7B | - | 5.12 | 63.6 | **60.6** | 74.8 | 60.8 | 31.3 | 61.0 | 58.7 |
| Pythia-2.8B (Biderman et al., 2023) | 2.8B | - | **5.04** | 64.7 | 59.3 | 74.0 | 64.1 | **32.9** | 59.7 | 59.1 |
| **Ours (TokenFormer-1.5B)** | 1.5B | 6.91 | 5.24 | 64.7 | 60.0 | **74.8** | 64.8 | 32.0 | 59.7 | 59.3 |

Table 10: (**Zero-shot Evaluations.**) The best results for each model size are highlighted in bold. We compare our models against open-source language models (LMs), including RNN-based methods, with various tokenizers, trained for up to 300B tokens.

Specifically, We begin by reformulating a two-layer MLP as an attention mechanism (Geva et al., 2021; Sukhbaatar et al., 2020), using this module as the foundational computational unit for neural network design, instead of the traditional linear projection, maximizing the flexibility of Transformers.

Based on this all-attention framework, we treat each key-value parameter pair as a learnable unit. By incrementally expanding this parameter set, the model grows progressively from a pre-trained smaller model to a larger one. This strategy achieves comparable performance to training a large model from scratch while greatly reducing computational costs.

The concept of tokenizing everything and leveraging attention for establishing connections is highly versatile. For example, it can be extended to the hidden states in linear models, allowing the memory to expand incrementally. This incremental memory expansion has the potential to mitigate excessive information loss, a common limitation in models such as Mamba (Gu & Dao, 2023) and RWKV (Peng et al., 2023).

## G.2   INCREASING INTERPRETABILITY

As TokenFormer is fully grounded in attention mechanisms, it leverages the inherent interpretability provided by attention, especially in token-parameter interactions. Geva et al. (2021) and Rigotti et al. (2021) studied Feed-Forward Networks (FFNs) and demonstrated that they could be interpreted as key-value memory structures. These memory structures encode learned patterns, which are selectively activated depending on the specific input. Our TokenFormer uses pattention as the most fundamental computational unit in the network design, rather than the less interpretable linear projection. Token-Parameter interactions provide an additional way of visualizing the interaction between the model parameters and the input via the attention score map, which may be helpful to further understand the behavior of the model.

## G.3   PRACTICAL LIMITATIONS

The complexity of token-token interactions is manageable, while token-parameter computations scale linearly with network size. This creates challenges in expanding parameters without increasing inference overhead. However, TokenFormer's inherent compatibility with sparse inference makes it an ideal candidate for integrating MoE techniques in future work, aiming to build more efficient and powerful networks. A remaining challenge is achieving efficient communication between experts when each key-value parameter token functions as an independent expert.

