# OpenReview forum: "TokenFormer: Rethinking Transformer Scaling with Tokenized Model Parameters"
_ICLR.cc/2025/Conference — ICLR 2025 Spotlight_

### Official Review · Reviewer_UJd1 · 2024-10-31

**Soundness:** 3
**Presentation:** 2
**Contribution:** 4
**Rating:** 8
**Confidence:** 4

**Summary:**

A novel growing architecture called "tokenformer" is introduced. The idea is to replace linear layers in the transformer architecture with a cross attention mechanism where input dependent queries cross attend to key and value parameters. This framing allows for the architecture to accommodate new parameters concatenated to the original key and value parameters. The architecture and training method are validated empirically by pretraining a 124M parameter tokenformer and incrementally growing it to 1.4B parameters. The resulting model performs competitively on evaluations compared to other models of similar size pretrained from scratch.

**Strengths:**

The main strength of the proposed method in the paper is the flexibility of the architecture to accommodate additional parameters without jeopardizing the capabilities of the original model. Consequently there is no additional cost to "healing" the performance of the original model before continually training and allowing the model to exploit the added parameters. The experimental results reported in the paper are compelling and the key ideas are explained adequately clearly.

**Weaknesses:**

The presentation of the mathematical form of the "Pattention" operation can be improved - specifically, the notation in equations 4 and 5 are confusing - the function \Theta is described as a "a modified softmax operation for stable optimization of Pattention layer" when in fact the function used in subsequent sections and experiments isn't really a softmax and what's more the function name \Theta doesn't appear. Also it is unclear how equation 5 connects to equation 4, i.e. what exactly the function f has to do with \Theta. Perhaps it would be better to remove any reference to \Theta, and to solely describe the operation in terms of the function f, and to make explicit the choice of the specific form of f clear. Ideally a glossary defining all mathematical symbols in the paper would really help.
 In fact, the authors should make it clear that the structure of the Pattention operation can be seen as an MLP with standard activations: GeLU(RMSNorm(XK)V). The analogy with attention is interesting, but it somewhat obscures the key mathematical form of the operation - especially since there are no tokens (i.e. data) involved in the K and V parameters in Pattention. Also it isn't made clear how the V parameters are expanded - are they also padded with 0?

Also, in addition to the comparison with the Net2Net method, have the authors considered comparing to say the method in  arXiv:2409.12903v2 [cs.CL]  which also aims directly at growing transformer models?

**Questions:**

Have the authors considered a linear attention version of Pattention, i.e. a version without the gelu activation in the cross attention? Also, it seems that the tokenformer replacement of linear layers can be made in any architecture, i.e. not just transformers. Have the authors experimented with alternative backbone architectures?

---

> ### Author Response · Authors · 2024-11-22
> **Response to Reviewer UJd1**
>
> We are glad the reviewer found the ideas in the paper compelling and clear. Below are our responses to the questions and suggestions raised by the reviewer.
>
> **Q1: The presentation of the mathematical form of the "Pattention" operation can be improved.**
>
> **A1:** Thank you for your feedback. We have removed the reference to $\Theta$ and added the notion table in the paper as requested. Please refer to lines 184-187 and lines 199-203 of the main paper for more details.
> ***
>
> **Q2: The authors should make it clear that the structure of the Pattention operation can be seen as an MLP with standard activations: GeLU(RMSNorm(XK)V). The analogy with attention is interesting, but it somewhat obscures the key mathematical form of the operation.**
>
> **A2:** Thanks. More discussions have been added to the main paper. Please refer to section G.1 in the appendix and to the general answer above. Our goal is to provide a unified framework for handling computations in models by treating parameters as tokens and using the attention mechanism to manage interactions between different types of tokens (e.g., data tokens and parameter tokens). We started by drawing an analogy between a two-layer MLP and attention, and used this module as the fundamental computational unit to design the model, rather than the previous linear projection, thereby unifying all the computations within a single framework. Then it was applied to incremental model scaling, enabling the expansion of a model based on a pre-trained one, significantly reducing training costs.
>
> Furthermore, the concept of tokenizing all components and leveraging attention to establish relationships has the potential to impact topics beyond pretraining models. For instance, it can be applied to the hidden states in linear models, allowing for incremental memory expansion while addressing the issue of excessive information loss often observed in such models.
> ***
>
> **Q3: It isn't made clear how the V parameters are expanded - are they also padded with 0?**
>
> **A3:** The value parameters are initialized randomly. More clarifications have been added to the main paper. Please see line 269 in the main paper.
> ***
>
> **Q4: In addition to the comparison with the Net2Net method, have the authors considered comparing to say the method in arXiv:2409.12903v2 [cs.CL] which also aims directly at growing transformer models?**
>
> **A4:** Thank you for pointing out this baseline. We implemented the method from this paper and conducted a preliminary experiment. The results are as follows:
>
> | Method    | model | 124M (60B) | 354M (15B)   | 757M (15B)  |
> |:----------------:|:----------------:|:------------------:|:------------:|:----------:|
> | | | PPL / Flops | PPL / Flops | PPL / Flops |
> | Net2Net        | Transformer | 16.4  &nbsp;/&nbsp; 3.6       | 14.3 &nbsp;/&nbsp; 6.8         |   12.1 &nbsp;/&nbsp; 10.1     |
> | HyperCloning (2024.09) | Transformer | 16.4 &nbsp;/&nbsp; 3.6       | 14.1 &nbsp;/&nbsp; 6.8         |   12.0 &nbsp;/&nbsp; 10.1     |
> | Ours           | TokenFormer | 16.1 &nbsp;/&nbsp; 3.6        | 13.7 &nbsp;/&nbsp; 3.6        |   11.5 &nbsp;/&nbsp; 3.6    |
>
> The unit of Flops is $10^4 \times T^2$. Our method outperforms HyperCloning while requiring less computational effort for token-token interactions. Due to time limitations, we scaled our model only up to 757M parameters. The complete comparison with this baseline will be included in the final camera-ready version.
> ***
>
> **Q5: Have the authors considered a linear attention version of Pattention, i.e. a version without the gelu activation in the cross attention?**
>
> **A5:** Thank you for this valuable suggestion. It highlights an excellent opportunity to further minimize the computational overhead associated with Token-Parameter interactions. As part of our future research, we will explore methods to reduce this cost even further, including techniques like Mixture-of-Experts (MoE) or linear attention.
> ***
>
> **Q6: Also, it seems that the tokenformer replacement of linear layers can be made in any architecture, i.e. not just transformers. Have the authors experimented with alternative backbone architectures?**
>
> **A6:** Thanks. Due to time constraints, we will explore additional backbone architectures in the future to broaden our approach. The concept of tokenization is not limited to parameters—it can also be applied to memory and hidden states. For instance, if the hidden states in models like Mamba or RNN are treated as tokens and processed using the attention mechanism, they can be expanded arbitrarily, thereby addressing the issue of excessive information compression in linear models.

---

> > ### Comment · Reviewer_UJd1 · 2024-11-26
> >
> > I thank the authors for their clarifications, I'm happy with the changes they made to their draft. In particular, I appreciate the comparison to the the Hypercloning method and in light of time, I find the results they reported to be adequate.

---

> ### Author Response · Authors · 2024-11-27
> **Official Comment by Authors**
>
> Thank you very much for your valuable suggestions and kind acknowledgment of our work! The complete experimental results of hypercloning (from 124M to 1.4B) will be included in the final version.

---

### Official Review · Reviewer_eA2h · 2024-11-04

**Soundness:** 4
**Presentation:** 4
**Contribution:** 3
**Rating:** 8
**Confidence:** 3

**Summary:**

The propsoed TokenFormer replaces dense linear layers of transformers with learnable token parameters to achieve progressive model scaling and reduce training cost compared with retraining from scatch of larger models with full data. The original matrix vector production is also replaced with the novel cross-attention between input token with tokenized parameters. Therefore, more token parameters can be easily added to well-trained smaller models incrementally due to flexibility of attention. The training cost is significantly reduced by 3x with similar or even better model performance. This paper proposes a interesting direction to scale LLMs and uses attention for almost all components in LLMs.

**Strengths:**

This paper proposes the novel TokenFormer that replaces the original linear layers in transformers with Pattention layers to enable more efficient and progressive model scaling without retraining from scratch. Pattention tokenizes the dense weights of query, key, value, and output layers with flexible and scalable token parameters. It then introduces cross-attention between input tokens with those token parameters. Instead of increasing the hidden dimension in transformers, TokenFormer achieves scaling by increasing the number of token parameters, incrementally adds new token parameters, and train all old and new token parameters with less training data. Therefore, it enables effectively reusing smaller model knowledge or weights and faster training of larger models with similar architecture. Experimental results show that the training cost of a 1.4B model is significantly reduced by training a 124M model initially and progressively scaling to 1.4B parameters, compared with training from scratch of a 1.4B model with 300B tokens.

**Weaknesses:**

1. In this paper, only results of scaling only up to 1.5B parameters are given. What are the complexity and effectiveness of scaling to larger scales such as 7B, 13B, 70B, or even more? In which case, the number of token parameters will increase multiple times and may be larger than token or model dimension.
2. The hidden dimension is relatively constrained to enable progressively scaling and reusing token parameters. Will it limit the capability of LLMs as FFN layers contain knowledge?

**Questions:**

same as weakness.

---

> ### Author Response · Authors · 2024-11-22
> **Response to Reviewer eA2h**
>
> We sincerely thank the reviewer for providing thoughtful review and positive feedback. Below are our responses to the questions and suggestions raised by the reviewer.
> ***
>
> **Q1: In this paper, only results of scaling only up to 1.5B parameters are given. What are the complexity and effectiveness of scaling to larger scales such as 7B, 13B, 70B, or even more? In which case, the number of token parameters will increase multiple times and may be larger than token or model dimension.**
>
> **A1:** Pretraining LLMs requires huge computational resources, and given our own resource limitations, the model in the paper is scaled up to 1.4B. We are also exploring scaling to larger models and integrating the MoE framework to design even bigger and more powerful models. The results of larger models will be included in camera ready version.
> ***
>
> **Q2: The hidden dimension is relatively constrained to enable progressively scaling and reusing token parameters. Will it limit the capability of LLMs as FFN layers contain knowledge?**
>
> **A2:** Our experiments have not observed a bottleneck in the hidden dimension. OpenAI's scaling laws [1] show that model capacity scales with size following a power law, with minimal impact from architectural details like width or depth.  Please see lines 3-4 of the abstract in [OpenAI's scaling laws paper](https://arxiv.org/pdf/2001.08361).
>
> Moreover, expanding parameters using the mixture-of-experts (MoE) approach while keeping the hidden dimension fixed is also a common industry practice [2,4]. TokenFormer is similar to this approach. This approach is validated by Krajewski et al. [3] and Fedus, William, et al [4], who found that increasing the number of experts while maintaining a fixed hidden dimension aligns with scaling laws for dense models. Please check Figure 1 of the popular [switch transformer](https://arxiv.org/pdf/2101.03961) for more details..
>
> [1] Kaplan, Jared, et al. "Scaling laws for neural language models." arXiv preprint arXiv:2001.08361 (2020).
>
> [2] Dai, Damai, et al. "Deepseekmoe: Towards ultimate expert specialization in mixture-of-experts language models." arXiv preprint arXiv:2401.06066 (2024).
>
> [3] Krajewski, Jakub, et al. "Scaling laws for fine-grained mixture of experts." ICLR 2024 Workshop Oral.
>
> [4] Fedus, William, Barret Zoph, and Noam Shazeer. "Switch transformers: Scaling to trillion parameter models with simple and efficient sparsity." Journal of Machine Learning Research 23.120 (2022): 1-39.

---

> > ### Author Response · Authors · 2024-12-04
> >
> > Thank you for reconsidering your evaluation of our paper. We truly appreciate your updated score and your constructive feedback on our work.

---

### Official Review · Reviewer_M5Lm · 2024-11-04

**Soundness:** 3
**Presentation:** 3
**Contribution:** 3
**Rating:** 6
**Confidence:** 4

**Summary:**

This paper proposes an innovative modification to the standard transformer architecture by replacing all linear layers with learned tokens via attention operations. By making this alteration, the proposed architecture can effectively remove all learned parameters (except embedding layers), instead replacing them with learned tokens. This token-parameter-attention mechanism (Pattention) therefore provides an efficient mechanism for scaling parameter counts. The authors validate their architecture with several scaling experiments, on both causal language modeling and vision tasks, demonstrating comparable performance with a significantly reduced training budget.

Overall, this paper makes a significant architectural contribution to the lineage of transformer architectures. However, the authors primarily focus on the empirical validation of their methodology without sufficient theoretical backing or exploring limitations of this approach. I am therefore inclined to reject, but would be willing to reconsider pending further discussion.

**Strengths:**

This paper proposes a unique approach to scaling parameters and demonstrates compelling results under those conditions, including vision and language tasks. The authors further support their claims with extensive experimentation, demonstrating a substantial reduction in training costs. Additionally, the authors propose an interesting attention mechanism for their parameter tokens, which solves the issue of vanishing gradients caused by the softmax function in traditional attention. Furthermore, this paper is well-written and easy to follow, with clear figures and tables.

**Weaknesses:**

-	Throughout the paper, the authors seem to imply that the scaling issue is unique to transformers. However, the increased training cost with scale applies to most DNN architectures. Furthermore, they contradict themselves when attributing training cost to increasing parameters while later showing that cost is dominated by self-attention (which is well known). The proposed training improvements are therefore solely due to fixing the hidden dimension size and resuming, rather than directly due to the Pattention mechanism. The authors should make their motivation more precise, and avoid overly vague and potentially misleading arguments to support their approach.
-	The paper lacks a theoretical foundation as to why their approach works, only briefly mentioning a potential explanation in Section 5, Future Work. Although scaling parameters generally improves performance, this is typically done while also scaling the hidden dimension, which has other benefits beyond parameter count (e.g. larger representation space). While the empirical results are promising, discussing deeper theoretical insights would allow application to other (hybrid) architectures beyond TokenFormer.
-	The authors compare against an old baseline (Net2Net) to demonstrate their model's effectiveness in growth when resuming from a smaller model. It would be more appropriate to compare with newer methods, which have been shown to perform much better (Yao et al., 2024). Furthermore, the practice of initializing extended queries to zero may negatively impact potential performance due to symmetry, where the proposed method instead relies entirely on the value token initialization. The authors should clarify why Net2Net was chosen over newer methods, or why the newer methods would not be a more suitable comparison.

**Questions:**

1. Your main argument for scaling is based on adding parameter tokens to the Pattention layers. However, this necessarily requires that you fix A) the hidden dimension, and B) the model depth.
    - 1a. Fixing the hidden dimension can lead to representation crowding, which typically does not occur when increasing it (e.g. larger vector space for representations). Could you investigate this with your scaled models? As a limitation it should result in an upper limit to capacity, where adding new parameter tokens will no longer improve performance.
    - 1b. Fixing the model depth can have a significant impact on the range of functions the transformer can learn. It’s possible that this was not apparent due to the model size and benchmarks, but would severely limit feasibility for more complex problems, which often require larger scales (>70B). Could you address this limitation?
    - 1c. By fixing the hidden dimension, achieving improved performance would then depend on expanding the representation span within a fixed vector space (i.e. approaching 100% channels to capture the variance). This will likely have a negative impact on quantization methods, which may offer substantial computational improvements beyond what Pattention can provide alone. Could you address this?
2. The parameter token scaling method raises several questions that should be addressed.
    - 2a. When scaling the number of parameter tokens, do you observe representation collapse, or do you apply a regularization term to prevent this? Adding fixed parameters without enforced symmetry breaking often leads to vector consolidation into tight groupings. Alternatively, have you confirmed that this behavior is not occurring?
    - 2b. In Figure 2, you suggest keeping the previously learned parameter tokens unfrozen for model growth; is this necessary? It may be more beneficial to keep the previously learned parameters frozen to essentially learn “residuals”.
3. While you present the mechanism by which Pattention works, there is no explicit connection to linear layers, or is it closer to an FFN? Could you clarify this connection in the appendix? Intuitively, if the GELU scoring is one-hot, then you have a K-V lookup.
4. Your scaling conclusions in Section 4.3, Table 3, and Figure 5 are contingent upon which term dominates in the asymptotic analysis. You require that $l \cdot n \cdot d \cdot T < l \cdot d \cdot T^2$, otherwise Pattention will become more expensive than simply using linear layers. The scaling argument should be more precise, and both acknowledge this point and demonstrate which regime the different model scales are in. Notably, a significant number of parameters were added to the 1.4B model (6.6M per matrix vs. 2.4M), where it would also be appropriate to report the FLOPs required for a forward pass of sequence length $T$, verified with empirical measurement. Furthermore, relying on a fixed $l$ while scaling $n$ may significantly impact parallelism during inference.
5. To confirm, did you retain the original softmax attention for the MHA layers, or did you apply the GELU scoring there as well? This should be made clear to avoid confusion.
6. Questions about the Future Work section. While it is understandable to suggest exploratory directions in the future work section, some of the proposed ideas would benefit from either preliminary results or a more concrete implementation plan.
    - 6a. You mention adjustable computational cost: can you elaborate on how this could be done? Your current methodology seems to be restricted to $n_\mathrm{new}$ once scaled up, without any way to “adjust” the computational cost.
    - 6b. The idea of combining vision and language parameter tokens is interesting; however, I am highly skeptical whether this would work. Do you have preliminary results to back up this idea? If not, perhaps the statement should be weakened, as it's not improbable that some features may have significant overlap, allowing for $\mathrm{max}(n_\mathrm{lang}, n_\mathrm{vision}) < n < n_\mathrm{lang} + n_\mathrm{vision}$, where the vision and language tasks are not simply combined after independent training.
    - 6c. The paragraph on “Increasing interpretability” is vague. How exactly will Pattention improve interpretability? Can you make a connection to Sparse Auto-encoders (SAEs)? This seems like something that should have been added to the appendix.




Additional Comments:

-	It appears you are missing a factor of 2 in the parameter counts in Table 3. According to your description, you learn $n$ Key and $n$ Value tokens, resulting in a total of $2n$. The parameter counts and cost would then be identical to a traditional transformer when $n=d_\mathrm{model}$ and $d_\mathrm{token}=d_\mathrm{model}$.
-	Please include a discussion connecting the GELU scoring function to your scaling ability. Scaling by setting new keys to zero relies on the GELU function, where GELU(0)=0; in contrast, exp(0)=1, which causes a norm shift.


Yao, Yiqun, Zheng Zhang, Jing Li, and Yequan Wang. ‘Masked Structural Growth for 2x Faster Language Model Pre-Training’. In The Twelfth International Conference on Learning Representations, 2024.

---

> ### Author Response · Authors · 2024-11-22
> **Response to Reviewer M5Lm (1/6)**
>
> We sincerely thank the reviewer for providing thoughtful review and constructive suggestions. We answer the questions as follows.
> ***
>
> **Q1: The practice of initializing extended keys to zero may negatively impact potential performance due to symmetry, where the proposed method instead relies entirely on the value token initialization.**
>
> **A1:** We thank the reviewer for raising an interesting point. Initializing the key parameter tokens to zero does not lead to symmetry issues. This approach is validated by the successful LoRA [1], where the first layer is also initialized to all zeros, while the second layer is initialized with Gaussian random values. This strategy minimizes noise during the initial training phase and preserves the original network's distribution. The symmetry problem can be avoided by ensuring the update process builds upon a pre-trained model and that the value parameter tokens are randomly initialized. Firstly, the random initialization of the value parameters adds diversity to the gradients. Secondly, the newly added parameters are incrementally updated on top of the well-trained base model. Consequently, the key parameters will quickly evolve into non-zero values during training.
>
> We further conducted validation experiments on OpenWebText2, scaling 124M model to 354M. The impact of different initializations and the final statistics of the key parameters in last Pattention layer are as follows:
>
> | key param    | value param    | test loss | std of old key | std of new key |
> |:----------------:|:------------------:|:------------:|:----------:|:----------:|
> | zero init        | random init        |   2.65     |  0.0437      |  0.0311 (0 $\rightarrow$ 0.0311)      |
> | random init      | random init        |   2.66     |  0.0438      |  0.0323  |
>
> As shown in the table above, even when the key parameter is initially set to zero, it quickly deviates from zero during training (increasing from 0 to a standard deviation of 0.0311 by the end of the process).
>
> [1] Hu, Edward J., et al. "Lora: Low-rank adaptation of large language models." ICLR2022

---

> ### Author Response · Authors · 2024-11-22
> **Response to Reviewer M5Lm (2/6)**
>
> **Q2: By fixing the hidden dimension, achieving improved performance would then depend on expanding the representation span within a fixed vector space (i.e. approaching 100\% channels to capture the variance). This will likely have a negative impact on quantization methods, which may offer substantial computational improvements beyond what Pattention can provide alone.**
>
> **A2:** Given a fixed number of model parameters, whether a larger representation space leads to greater expressive power remains an open problem. In our experiments, we did not encounter the bottlenecks in the hidden dimension. With the same data and training budget, OpenAI's scaling laws [1] reveal that **model capacity follows a power-law relationship with model size. The architectural details, such as network width or depth, have minimal impact on this relationship**. Please see lines 3-4 of the abstract in [OpenAI's scaling laws paper](https://arxiv.org/pdf/2001.08361).
>
> Additionally, expanding model parameters through the mixture-of-experts (MoE) approach—where the hidden dimension remains fixed while more experts are added—has become a popular and successful practice in the industry [2,3]. TokenFormer represents a variant of MoE. Fedus, William, et al [2] and Krajewski, Jakub, et al. [4] have demonstrated that by gradually increasing the number of experts without changing the hidden dimension, models can adhere to the scaling laws corresponding to dense models. This experimental conclusion validates the feasibility of our method. Please check Figure 1 of the popular [switch transformer](https://arxiv.org/pdf/2101.03961) for more details.
>
> Notably, our approach does not conflict with methods that expand the hidden dimension; instead, we provide an alternative scaling dimension beyond the hidden dimension. In Transformers, the qkv (query, key, value) and output projection matrices are square, meaning the channel and hidden dimensions are linked. Given a fixed network depth, this linkage forces Transformers to scale the model only by increasing the hidden dimension, which significantly raises inference costs. Such scaling is unsustainable in practical applications with limited computational resources. Our approach offers a new scaling dimension that is separate from the hidden dimension, helping to keep token-token interaction costs manageable when modeling long sequences.
>
> [1] Kaplan, Jared, et al. "Scaling laws for neural language models." arXiv preprint arXiv:2001.08361 (2020).
>
> [2] Fedus, William, Barret Zoph, and Noam Shazeer. "Switch transformers: Scaling to trillion parameter models with simple and efficient sparsity." Journal of Machine Learning Research 23.120 (2022): 1-39.
>
> [3] Dai, Damai, et al. "Deepseekmoe: Towards ultimate expert specialization in mixture-of-experts language models." arXiv preprint arXiv:2401.06066 (2024).
>
> [4] Krajewski, Jakub, et al. "Scaling laws for fine-grained mixture of experts." ICLR 2024 Workshop Oral.
> ***
>
> **Q3: The paper lacks a theoretical foundation as to why their approach works, only briefly mentioning a potential explanation in Section 5, Future Work. While the empirical results are promising, discussing deeper theoretical insights would allow application to other (hybrid) architectures beyond TokenFormer.**
>
> **A3:** The theoretical analysis of LoRA's representational capacity [2] can provide theoretical insights into our method to some extent. As described in Equations 4-13 of main paper, given input $X$ and learnable tokens $(K_P, V_P)$, we formulate token-parameter interaction as attention, $\Theta\left(X \cdot K_P^{\top}\right) \cdot V_P$. With this naturally expandable design, the model can be incrementally scaled by appending new key-value parameter tokens, $\Theta\left(X \cdot [K_P^{\text{old}} , K_P^{\text{new}}]^{\top}\right) \cdot [V_P^{\text{old}}, V_P^{\text{new}}]$. The effectiveness of this approach can be partly understood through the analysis of LoRA [1], as TokenFormer can be viewed as a high-rank extension of LoRA, where $K_P^{\text{new}}$ and $V_P^{\text{new}}$ are equal to $A$ and $B$ matrix. LoRA has been widely applied in foundation models, and its representational capacity has been theoretically proven by Zeng Y et al [2], which to some extent explains why TokenFormer’s incremental scaling can be effective. This preliminary analysis has been added to the main paper. Please see G.4 in the appendix for more details.
>
> [1] Hu, Edward J., et al. "Lora: Low-rank adaptation of large language models." ICLR2022
>
> [2] Zeng, Yuchen, and Kangwook Lee. "The expressive power of low-rank adaptation." ICLR2024

---

> ### Author Response · Authors · 2024-11-22
> **Response to Reviewer M5Lm (3/6)**
>
> **Q4: The authors compare against an old baseline (Net2Net) to demonstrate their model's effectiveness in growth when resuming from a smaller model. It would be more appropriate to compare with newer methods, which have been shown to perform much better (Yao et al., 2024). The authors should clarify why Net2Net was chosen over newer methods, or why the newer methods would not be a more suitable comparison.**
>
> **A4:** Net2Net is a widely used method in the field of model reuse, valued for its simplicity and efficiency. It remains the most common strategy in the industry [1,2], requiring no extra loss functions or operations—only using parts of a smaller model to initialize a larger one. This simplicity makes it our primary baseline.
>
> To better validate the effectiveness of our method, we included HyperCloning [1] as an additional baseline. This is the most recent paper on model scaling (published two months ago), and it was also suggested by Reviewer UJD1 as a strong baseline. All experiments were conducted on OpenWebText2, and the specific perplexities are as follows:
>
> | Method    | model | 124M (60B) | 354M (15B)   | 757M (15B)  |
> |:----------------:|:----------------:|:------------------:|:------------:|:----------:|
> | | | PPL / Flops | PPL / Flops | PPL / Flops |
> | Net2Net        | Transformer | 16.4  &nbsp;/&nbsp; 3.6       | 14.3 &nbsp;/&nbsp; 6.8         |   12.1 &nbsp;/&nbsp; 10.1     |
> | HyperCloning (2024.09) | Transformer | 16.4 &nbsp;/&nbsp; 3.6       | 14.1 &nbsp;/&nbsp; 6.8         |   12.0 &nbsp;/&nbsp; 10.1     |
> | Ours           | TokenFormer | 16.1 &nbsp;/&nbsp; 3.6        | 13.7 &nbsp;/&nbsp; 3.6        |   11.5 &nbsp;/&nbsp; 3.6    |
>
> The unit of Flops is $10^4 \times T^2$. Our method still outperforms the latest HyperCloning and has lower computational complexity in the token-token interaction part. More baselines will be included in the camera-ready version.
>
> [1] Samragh, Mohammad, et al. "Scaling Smart: Accelerating Large Language Model Pre-training with Small Model Initialization." arXiv preprint arXiv:2409.12903 (2024).
>
> [2] Wang, Peihao, Rameswar Panda, and Zhangyang Wang. "Data efficient neural scaling law via model reusing." ICML, 2023.
> ***
>
> **Q5: Fixing the model depth can have a significant impact on the range of functions the transformer can learn. It’s possible that this was not apparent due to the model size and benchmarks, but would severely limit feasibility for more complex problems, which often require larger scales (>70B). Could you address this limitation?**
>
> **A5:** Simply scaling the model by increasing its depth is unsustainable. Very deep networks often suffer from gradient vanishing or exploding problems, which make training exceedingly difficult. Consequently, even X-AI's [Grok-1](https://github.com/xai-org/grok-1) model with 314 billion parameters is limited to just 64 layers. We offer an alternative dimension to scaling the model beyond simply increasing its depth.
>
> Notably, our approach does not conflict with methods that stack the depth. TokenFormer can also be scaled in depth by initializing new layers on top of the base model to further expand the model depth.
>
> |method | hidden dim   | parameter tokens   | layer | param amount |final loss |
> |:----------------:|:------------------:|:------------:|:----------:|:----------:|:----------:|
> |base model | 768        | 576        | 12         |  124M | 2.82      |
> |add param tokens | 768        | 2140 (+1564)        | 12 (+0)         | 354M | 2.65      |
> |add param tokens + depth| 768        | 1072 (+496)        | 24 (+12)  |354M       |  2.64  |
>
> The table shows that TokenFormer can not only scale the model by increasing the number of parameter tokens but also be compatible with depth expansion.

---

> ### Author Response · Authors · 2024-11-22
> **Response to Reviewer M5Lm (4/6)**
>
> **Q6: When scaling the number of parameter tokens, do you observe representation collapse, or do you apply a regularization term to prevent this? Adding fixed parameters without enforced symmetry breaking often leads to vector consolidation into tight groupings. Alternatively, have you confirmed that this behavior is not occurring?**
>
> **A6:** We did not observe representation collapse because our method of adding parameters is similar to LoRA and does not present symmetry issues. Similar to LoRA, our approach builds on the original model, with value parameters randomly initialized. This provides diverse gradients for different parameter tokens, effectively avoiding symmetry issues. Even if the key parameters are initially set to all zeros, by the end of training, the norm of the new key parameters is on the same order of magnitude as the old parameters.
> ***
>
> **Q7: Throughout the paper, the authors seem to imply that the scaling issue is unique to transformers. However, the increased training cost with scale applies to most DNN architectures.**
>
> **A7:** Currently, the Transformer is the dominant general-purpose architecture due to its flexibility and versatility. It has achieved remarkable success across various domains, particularly in NLP, where model scaling is highly emphasized. Therefore, we focus primarily on the Transformer and push its flexibility to the limit. However, the concept of tokenizing everything is not exclusive to Transformers and can also be
> applied to linear models, like Mamba and RWKV. Specifically, if the hidden states in linear models are treated as tokens and processed by attention, they can be expanded arbitrarily, thereby having the potential to address the problems of excessive information compression.
>
> ***
>
> **Q8: Furthermore, they contradict themselves when attributing training cost to increasing parameters while later showing that cost is dominated by self-attention (which is well known).**
>
> **A8:** Whether Token-Token or Token-Parameter interactions dominate the computational overhead depends on the sequence length. TokenFormer effectively reduces the computational cost of both. Research on model reuse and MoE has targeted reducing Token-Parameter interaction costs during training and inference, while methods like Mamba and linear Transformers focus on optimizing Token-Token interaction costs. Our TokenFormer combines the strengths of these two research directions, enabling incremental model reusing with potential applications in sparse inference. Additionally, it offers the advantage of controllable Token-Token Interaction costs, making it well-suited for long-text modeling tasks.
> ***
>
> **Q9: The proposed training improvements are therefore solely due to fixing the hidden dimension size and resuming, rather than directly due to the Pattention mechanism. The authors should make their motivation more precise, and avoid overly vague and potentially misleading arguments to support their approach.**
>
> **A9:** We use an attention-like mechanism to model Token-Parameter interaction, replacing linear projection. This enables the network’s Token-Parameter interaction to handle a variable number of parameters, similar to how attention processes a variable number of input tokens. Based on this design, we incrementally scale the model by adding new parameter tokens to the original one and initializing key parameters to 0, allowing us to fully resume training from pre-trained checkpoints — a capability not possible with transformers.
> ***
>
> **Q10: In Figure 2, you suggest keeping the previously learned parameter tokens unfrozen for model growth; is this necessary? It may be more beneficial to keep the previously learned parameters frozen to essentially learn “residuals”.**
>
> **A10:** In our experiments, we found that freezing old parameters leads to poorer performance, as shown in the table below:
>
> |Scaling setting  | Old Param    | New Param |  Perplexity |
> |:---------------:|:------------:|:---------:|:-----------:|
> |124M to 354M     | frozen       | unfrozen  | 14.2        |
> |124M to 354M     | unfrozen     | unfrozen  | 13.7        |
>
> Freezing old parameters will reduce the effective training budget, leading to poorer performance under the same number of training iterations.
> ***
>
> **Q11: While you present the mechanism by which Pattention works, there is no explicit connection to linear layers, or is it closer to an FFN? Could you clarify this connection in the appendix? Intuitively, if the GELU scoring is one-hot, then you have a K-V lookup.**
>
> **A11:** Thanks. The discussion has been added to the main paper. Please see the sections G.1 and G.2 in the appendix.
> ***

---

> ### Author Response · Authors · 2024-11-22
> **Response to Reviewer M5Lm (5/6)**
>
> **Q12: Your scaling conclusions in Section 4.3, Table 3, and Figure 5 are contingent upon which term dominates in the asymptotic analysis. You require that $l \cdot n \cdot d \cdot T < l \cdot d \cdot T^2$, otherwise Pattention will become more expensive than simply using linear layers. The scaling argument should be more precise, and both acknowledge this point and demonstrate which regime the different model scales are in. Notably, a significant number of parameters were added to the 1.4B model (6.6M per matrix vs. 2.4M), where it would also be appropriate to report the FLOPs required for a forward pass of sequence length $T$
> , verified with empirical measurement.**
>
> **A12:** We would like to clarify a potential misunderstanding. The terms $l \cdot n \cdot d \cdot T $ and $ l \cdot d \cdot T^2$ determine whether Token-Token or Token-Parameter computation dominates, which is irrelevant to whether pattention or linear layers have a higher computational cost. For the token-parameter part, the computational overhead is proportional to the number of network parameters, so it is the same whether pattention or linear layers are used. However, since we can scale the parameter size without increasing the hidden dimension, pattention significantly reduces the computational overhead for token-token interaction compared to a linear layer. As a result, compared to linear layers, the overall computational overhead is greatly reduced in long context modeling.
> ***
>
> **Q13. Furthermore, relying on a fixed $l$ while scaling $n$ may significantly impact parallelism during inference.**
>
> **A13:** In our experiments, we did not observe this issue. The specific training TPU hours are shown in the table below.
>
> |Model  | #Params    | Layer |  hidden dimension | Parameter Tokens | TPU hours |
> |:---------------:|:------------:|:---------:|:-----------:|:-----------:|:-----------:|
> |Transformer     | 354M     | 24  | 1024       | -         |  421|
> |Tokenformer     | 354M     | 12  | 768       | 2140       |  381|
> |Transformer     | 757M     | 24  | 1536       | -         |  741|
> |Tokenformer     | 757M     | 12  | 768       | 4850       |  689|
> |Transformer     | 1.3B     | 24  | 2048       | -         | 1163|
> |Tokenformer     | 1.3B     | 12  | 768       | 8620       | 1149|
>
> All the models are trained on 30B tokens with a sequence of 1024 on TPU v4 hardware. Since the sequence length here is 1024, the computational cost is primarily in the Token-Parameter interaction, allowing for an approximate comparison of the issues related to matrix parallelization. The table shows that cost scales with parameter count and is unrelated to the matrix shape. Therefore, fixing the hidden dimension and increasing the number of parameter tokens won't impact parallelization.
> ***
>
> **Q14: To confirm, did you retain the original softmax attention for the MHA layers, or did you apply the GELU scoring there as well? This should be made clear to avoid confusion.**
>
> **A14:** Thanks. As outlined in Equations 8–10, we replaced all linear projections with pattention while preserving the standard attention mechanism for token-token interactions. This means that the standard softmax operation remains unchanged.
> ***
>
> **Q15. You mention adjustable computational cost: can you elaborate on how this could be done? Your current methodology seems to be restricted to $n_{\text{new}}$ once scaled up, without any way to “adjust” the computational cost.**
>
> **A15:** Unlike Transformers, which can only scale by increasing the hidden dimension at a given network depth, our approach decouples the hidden dimension from the model size and offers a new scaling dimension: the number of parameter tokens. Thanks to this new axis, the hidden dimension can be freely configured regardless of model size, thereby enabling adjustable control over the cost of token-token interactions.
>
> ***

---

> ### Author Response · Authors · 2024-11-22
> **Response to Reviewer M5Lm (6/6)**
>
> **Q16. The idea of combining vision and language parameter tokens is interesting; however, I am highly skeptical whether this would work. Do you have preliminary results to back up this idea? If not, perhaps the statement should be weakened, as it's not improbable that some features may have significant overlap, allowing for $\text{max}(n_{\text{lang}}, n_{\text{vision}}) < n < n_{\text{lang}} + n_{\text{vision}}$, where the vision and language tasks are not simply combined after independent training.**
>
> **A16:** We have conducted preliminary experiments on vision-language tasks, and the results are promising. Our approach introduces bridging parameter tokens to connect the vision and language parameters. Specifically, image tokens are allowed to attend only to image parameter tokens and bridging parameter tokens, while language tokens interact exclusively with language parameter tokens and bridging parameter tokens. However, tokens from all modalities can interact freely in the token-token attention mechanism.
>
> The models used include a TokenFormer pre-trained on ImageNet for vision modality and another TokenFormer pre-trained on OpenWebText2 for language modality, both at the base model size. For validation, we adopted the widely used BLIP's training data settings. The COCO captioning results are summarized below:
>
> | method    | vision model    | language model   | new param | Blue4/CIDEr |
> |:----------------:|:------------------:|:------------:|:----------:|:----------:|
> | BLIP        | ViT (87M,ImageNet)        | Bert (110M)         |   110M     |  38.6 / 128.7      |
> | Ours        | TokenFormer (87M, ImageNet) | TokenFormer (124M)  | 96M | 39.0 / 128.4     |
>
> The table above shows that our TokenFormer can perform vision-language tasks and achieve performance comparable to classic vision-language methods.
> ***
>
> **Q17. The paragraph on “Increasing interpretability” is vague. How exactly will Pattention improve interpretability? Can you make a connection to Sparse Auto-encoders (SAEs)? This seems like something that should have been added to the appendix.**
>
> **A17:** The interpretability discussed here is not related to sparse autoencoders but rather to research analyzing the interpretability of Feed-Forward Networks (FFNs). Geva, Mor, et al. [1] modeled FFNs as key-value lookup mechanisms, interpreting FFNs in Transformers as key-value memory structures where each key-value pair encodes a learned pattern. Different tasks activate and utilize these key-value memories in unique ways. This discussion has already been incorporated into the paper. Please see section G.2 in the appendix.
>
> [1] Geva, Mor, et al. "Transformer feed-forward layers are key-value memories." EMNLP (2021).
> ***
>
> **Q18. It appears you are missing a factor of 2 in the parameter counts in Table 3. According to your description, you learn $n$ Key and $n$ Value tokens, resulting in a total of $2n$. The parameter counts and cost would then be identical to a traditional transformer when $n=d_{\text{model}}$ and $d_{\text{token}} = d_{\text{model}}$.**
>
> **A18:** Sorry for this typo. We have already corrected it in the main paper. Please refer to the updated Table 3 in the main paper. The computational overhead of Token-Parameter interactions is strictly proportional to the number of network parameters. Therefore, when the parameter count is kept the same, the computational cost for this part is identical between TokenFormer and Transformer. However, our method decouples the token dimension from the model dimension, allowing the computational cost of Token-Token interactions to be independent of the model's parameter count. This ensures that scaling the model does not increase the computational burden of Token-Token interactions.
> ***
>
> **Q19. Please include a discussion connecting the GELU scoring function to your scaling ability. Scaling by setting new keys to zero relies on the GELU function, where GELU(0)=0; in contrast, exp(0)=1, which causes a norm shift.**
>
> **A19:** Thanks to the reviewer for raising that interesting observation. By replacing the exponential function in softmax with the GeLU function, we can add new parameters while fully resuming the checkpoint of the previous model. For the exponential function, if $\text{exp}(x) = 0$, $x$ would need to be negative infinity. It is not feasible to achieve this property by initializing the key parameters, making it impossible to ensure that newly added key parameters have no impact on the original key parameters. This is one advantage of using GeLU. More clarification has been added to section B (lines 769-773) in the appendix.

---

> > ### Comment · Reviewer_M5Lm · 2024-11-26
> > **Response to Authors (1/2)**
> >
> > I appreciate the authors’ detailed response and the additional experiments provided during the discussion phase. However, I remain concerned about the theoretical backing of the proposed method and the appropriateness of the comparisons drawn, particularly with transformer scaling under Net2Net.
> >
> > My key points of concern are as follows:
> >
> > 1. **Theoretical Motivation in the Main Paper:** While the authors have included some theoretical justification in the appendix, I believe this should be included in the main paper. A clear theoretical motivation is essential to understand the method’s applicability, rather than framing it solely as an architectural trick.
> >
> > 2. **Inconsistencies in Theoretical Analogies:** The authors’ response draws analogies between pattention and methods like LoRAs, MoEs, and FFNs. While these methods share some conceptual similarities, they are not equivalent, and the analogy lacks the clarity needed to provide a fundamental insight. This further underscores the absence of a solid theoretical underpinning, as raised in point 1.
> >
> > 3. **Contradictory Evidence from Additional Experiments:** The experiments provided appear to conflict with the LoRA analogy. In LoRA, base parameters are frozen, and a residual is learned. However, pattention does not freeze existing parameter tokens, which weakens this analogy. Additionally, LoRAs are typically applied to individual linear layers rather than entire FFNs. The observed connection to LoRAs appears to stem more from the use of zero initialization - a method employed by LoRAs and widely used to stabilize training dynamics generally - rather than from LoRA’s core principles.
> >
> > 4. **Implications for Baseline Comparisons:** Based on the discussion, pattention seems more analogous to an FFN, implying that every linear layer in the transformer is effectively replaced by an FFN. Under these conditions, comparing scaling results to a transformer baseline may introduce bias favoring TokenFormer. A fairer comparison might involve replacing the corresponding layers in the baseline with equivalently sized FFNs.
> >
> > ---
> >
> > ## Specific Responses:
> >
> > **A1.** Thank you for running this experiment. It suggests that zero initialization does not negatively affect performance, which is helpful to clarify.
> >
> > ---
> >
> > **A2.** While this may hold true for your experiments, it does not rule out the possibility that such behavior occurs under different conditions. It is plausible that your models have not yet hit their capacity, or that pattention increases capacity by replacing the linear layers, thereby shifting the saturation point further. While difficult to substantiate, if true, this would provide a compounding benefit of the pattention mechanism.
> >
> > I appreciate the authors’ effort to relate their methodology to best practices in MoE methods. However, I do not find the comparison fully justified. MoE models scale the hidden dimension of the base model and then hold this hyperparameter fixed while increasing the number of experts. In contrast, scaling TokenFormer is more akin to scaling the base model itself, rather than employing expert scaling.
> >
> > Additionally, I agree that scaling sequence length while keeping the hidden dimension fixed can help manage token-token interaction costs, particularly in tasks involving long sequences. However, in non-autoregressive models (e.g. ViT), scaling the hidden dimension is often more practical than scaling the sequence length. While inference costs are a valid concern, these trade-offs are highly context-dependent and should be carefully framed.
> >
> > ---
> >
> > **A4.** Thank you for providing this additional comparison. However, I find the results inconclusive due to potential confounding factors. Replacing linear layers with FFNs could inherently enhance performance, making it challenging to attribute superior scaling solely to the addition of parameter tokens. To draw definitive conclusions, it is essential to isolate this effect.
> >
> > ---
> >
> > **A5.** I agree that simply scaling by depth is unsustainable and can lead to stability issues, as you have pointed out. However, I believe this is not an effective argument for TokenFormer, as comparable models (smaller than Grok-1) do not encounter such limitations. Furthermore, it is well known that transformer depth influences the range of computable functions and tasks (Merrill et al., 2024), which could reduce the efficacy of TokenFormer without a comparable solution.
> >
> > That said, your additional experiment suggests that this limitation may not apply to TokenFormer. Including a comparison with TokenFormer initialized to the expanded configuration (e.g. n=1072, d=768, L=24) would provide further clarity. Specifically, it would help confirm whether the presented scaling results genuinely reflect comparable performance or are influenced by the scaling strategy (e.g., where the extra layers were added).
> >
> > Merrill et al. ‘The Illusion of State in State-Space Models’. (2024) arXiv:2404.08819

---

> > > ### Comment · Reviewer_M5Lm · 2024-11-26
> > > **Response to Authors (2/2)**
> > >
> > > ---
> > >
> > > **A6.** Could you provide empirical evidence to support this claim? Given the routing-like behavior of the pattention layers, measuring the routing probability density over representative batches would serve as an appropriate indicator. Additionally, evaluating the strength of the routing choice (e.g., via entropy) could offer further insight.
> > >
> > > ---
> > >
> > > **A8.** There may have been a misunderstanding regarding my concern. While the authors acknowledge that the dominant computational component depends on the sequence length, my question was aimed at establishing a more precise scaling argument, rather than presuming sequence length always dominates. This presumption may not hold true for vision transformers.
> > >
> > > I acknowledge that TokenFormer can reduce computational costs in both regimes; however, in the FFN-dominant regime, the computational cost will depend on the value of $n$. Specifically, when $d_0\cdot n > d^2$, TokenFormer could become more computationally expensive. Further clarity on this point would help solidify the scaling argument.
> > >
> > > ---
> > >
> > > **A9.** It seems that my concern may not have been fully understood. My question was focused on clarifying whether the observed training improvements are solely due to fixing the hidden dimension size and resuming, rather than being directly attributable to the pattention mechanism itself. This distinction should be stated more clearly.
> > >
> > > Notably, if the hidden dimension were also scaled, albeit less aggressively, TokenFormer might no longer exhibit such strong training improvements. Providing evidence to disentangle these effects would make the motivation and benefits of the pattention mechanism more precise.
> > >
> > > ---
> > >
> > > **A10.** This result fundamentally undermines the analogy to LoRA. LoRA relies on residual learning, where the base model remains frozen. In contrast, TokenFormer requires the base model to be unfrozen to achieve good results, breaking this key property. This distinction suggests that the analogy to LoRA may not be appropriate.
> > >
> > > ---
> > >
> > > **A12.** It seems there may have been a misunderstanding regarding my comment. I was referring to the last row in the table, where the dominant term is extracted. In this case, identifying which term is dominant is critical, as it impacts the relevance of the stated improvement of pattention over linear layers. Notably, when $n=d$, there is no improvement.
> > >
> > > This also suggests that the scaling comparison in Figure 5 may not be particularly meaningful, as $d$ and $l$ for the transformer model could be adjusted to match TokenFormer.
> > >
> > > ---
> > >
> > > **A13.** Thank you for your response. However, my concern was specifically about *inference*, not training. The reported training TPU hours do not necessarily reflect inference performance, where additional overheads can make parallelism harder to extract. That said, carefully evaluating inference scalability may indeed prove challenging, especially given the current model scales. As such, I consider this a minor issue.
> > >
> > > ---
> > >
> > > **A15.** It seems my comment may have been misunderstood. By scaling up to $n_{new}$, TokenFormer loses the ability to scale back down to $n < n_{new}$. While this can be achieved at discrete intervals through progressive training, I believe this approach is not fully aligned with the phrase “adjust computational costs.” In contrast, MoE models successfully achieve this flexibility by dynamically adjusting the number of routed experts.
> > >
> > > ---
> > >
> > > **A16.** This is very interesting; can these results be included in the appendix? Additionally, were the vision and language tokens frozen or unfrozen during the experiment?
> > >
> > > ---
> > >
> > > **A17.** Thank you for clarifying this point. Please include a reference to Appendix G.2 at the relevant point in Section 5.
> > >
> > > Additionally, in Appendix G.2, you assert that pattention “ensures that all Token-Parameter interactions … are interpretable”. This claim is unproven and remains speculative. It is also plausible that pattention layers may not achieve the same level of interpretability as traditional FFNs.
> > >
> > > ---
> > >
> > > **A18.** Thank you for addressing this typo. However, it appears that a factor of 2 is still missing in the last row.
> > >
> > > ---
> > >
> > > Minor comments with the revision:
> > > - The term “improved softmax” may be too confusing and implying that you are improving the softmax in the Token-Token attention mechanism as well. It may be worth considering an alternative name for this method to avoid confusion.
> > > - The linewidth in the legends for Figure 3 and 4 should be increased. It is difficult to tell which color is which.

---

> > > > ### Author Response · Authors · 2024-12-01
> > > > **Response to Reviewer M5Lm (6/6)**
> > > >
> > > > **A13:** We thank the reviewer for the understanding. For autoregressive models, training time and inference time are often correlated. However, to address the reviewer's concerns, we provide the inference time for TokenFormer and Transformer in an autoregressive manner, showing the time required for 100 steps (batch size=1, with KV cache, TPU-v3). While TokenFormer scales the number of parameter tokens and Transformer scales the hidden dimension, their wall clock time at inference time is similar under parallelism.
> > > >
> > > > |Model  | #Params    | Layer |  hidden dimension | Parameter Tokens | TPU hours | Inference Time (ms) |
> > > > |:---------------:|:------------:|:---------:|:-----------:|:-----------:|:-----------:|:-----------:|
> > > > |Transformer     | 354M     | 24  | 1024       | -         |  421| 1614 |
> > > > |Tokenformer     | 354M     | 12  | 768       | 2140       |  381| 1347 |
> > > > |Transformer     | 757M     | 24  | 1536       | -         |  741| 2622 |
> > > > |Tokenformer     | 757M     | 12  | 768       | 4850       |  689| 2309 |
> > > > |Transformer     | 1.3B     | 24  | 2048       | -         | 1163| 3913 |
> > > > |Tokenformer     | 1.3B     | 12  | 768       | 8620       | 1149| 3926 |
> > > >
> > > > **A15:** Thank the reviewer for raising this interesting points. We agree that the phrase "adjustable computational costs" is not very accurate, as our experiments primarily demonstrate the impact of increasing $n$. To solve this concerns, we have removed this statement in the revised version. However, it is worth noting that if the reviewer believes MoE can achieve this, then TokenFormer also has the potential to accomplish this goal by dynamically adjusting the number of routed experts (key-value parameter pair).
> > > >
> > > > **A16:** Thanks the reviewer for showing interest in the application of TokenFormer in vision-language tasks. The results presented are preliminary. We are actively advancing research in this direction to improve our model's performance in vision-language tasks and to benchmark it against the LLaVA method [1]. More comprehensive results and implementation details will be included in the appendix of the camera-ready version. In our method, the visual pre-trained parameter tokens and language pre-trained parameter tokens are both frozen, following the same strategy as the classic LLaVA approach (freezing visual encoder and language decoder).
> > > >
> > > > [1] Liu, Haotian, et al. "Visual instruction tuning." Advances in neural information processing systems 36 (2024).
> > > >
> > > > **A17:** We have included the reference (please check the line 530 in the revised manuscript). To address the concern of reviewer about over claim, we have modified this sentence "ensures that all Token-Parameter interactions … are interpretable" into "Token-Parameter interactions provide an additional way of visualizing the interaction between the model parameters and the input via the attention score map, which may be helpful to further understand the behavior of the model."
> > > >
> > > > **A18:** Thank the reviewer for catching this error. We have fixed it. Please see Table 3 in revision.
> > > >
> > > > **Minor:**
> > > > - We have revised "improved softmax" to "modified softmax."
> > > > - Thanks, we have fixed it. Please see the revised version.

---

> > > > > ### Comment · Reviewer_M5Lm · 2024-12-01
> > > > > **Response to Authors**
> > > > >
> > > > > Thank you for your detailed response and for performing the additional experiments. The results are indeed promising and significantly strengthen the contribution of Tokenformer.
> > > > > In the interest of time, I will address the immediate follow-up points now and respond to the remaining issues within the next day.
> > > > >
> > > > > ---
> > > > >
> > > > > **Comparison with LoRA:**
> > > > >
> > > > > Thank you for addressing this point. However, I believe there may be some confusion regarding my concern. LoRA learns a residual added to a frozen baseline matrix, ensuring only the residual is trained. In your approach, both the original parameters and the additional ones are trained together, effectively redistributing the learned representation across the expanded parameter space. This method is valid, but it differs fundamentally from LoRA’s residual-based framework, as unfreezing the original weights changes their role and blends them with the new parameters, making the approach closer to fine-tuning an expanded matrix than learning a residual.
> > > > >
> > > > > Notably, the LoRA analogy would be valid if your scaling method froze the previously learned parameters, treating the new ones as “residuals.” The results from A16, suggest that this approach is possible and potentially advantageous for fine-tuning, as it may be more efficient and scalable than LoRA - though I suspect this would require further careful investigation.
> > > > >
> > > > > ---
> > > > >
> > > > > **A6:** I presume this means the modified softmax is no longer normalized, which could introduce additional complications, particularly regarding interpretability. This redistribution to slightly lower values is interesting and aligns with my expectation, as the larger space likely spreads the distribution across the added tokens.
> > > > >
> > > > > However, the distribution you provided may be skewed by the lack of normalization. Could you renormalize along the parameter token dimension (after squaring) before computing the mean? Additionally, another way to check for collapse is to consider the histogram of top-1 tokens (after squaring) and compute the normalized entropy over that distribution: $\mathrm{diversity}=H/\mathrm{log}_2(N)$, where a uniform distribution would yield a value of 1.
> > > > >
> > > > > ---
> > > > >
> > > > > **A15:** Thank you for making this change. The statement regarding reducing top-k routing would hold if the model could still perform well with fewer parameter tokens after expansion. In this case, the top-k selection would effectively act as an adjustable mask. However, by unfreezing the tokens learned in the previous scale, performance would likely deteriorate below that of the original model if such a reduction were attempted. That said, I see an opportunity for Tokenformer to incorporate elements of both ideas, though without a concrete implementation, this remains speculative and may be better suited for future work.

---

> ### Author Response · Authors · 2024-12-01
> **Response to Reviewer M5Lm (1/6)**
>
> We sincerely thank the reviewer for their detailed feedback and specific suggestions. Below, we address the key concerns:
>
> **Theoretical Motivation in the Main Paper:**
> We appreciate the reviewer’s valuable suggestions. In the current version of the paper, this analysis has been included in the appendix due to its length and the space constraints of the main text. Incorporating it into the main paper within the page limit would require extensive rewriting. Due to the deadline for PDF revision, we assure the reviewer that we will thoroughly revise and integrate this theoretical analysis into the main text for the camera-ready submission.
>
> Our paper’s key contribution is the proposal of a novel architecture for foundation models, validated through extensive experiments. This architecture is inherently scalable and reduces training costs by reusing parameters. Notably, the theory of deep network remains an open problem. While more solid theoretical backing would undoubtedly strengthen our paper, we believe this is a promising direction for long-term exploration in the future.
>
> **Inconsistencies in Theoretical Analogies:**
> Our pattention is indeed different from LoRAs, MoEs, and FFNs, both in theory and in specific implementation. This is why we use the term "analogy". This approximate analogy is interesting (commented by reviewer UJd1) and helps readers understand at a high level why the pattention mechanism is effective. To address the reviewer's concern, the analogy of LoRA has been removed (G.4). Please check the revision.
>
> **Contradictory Evidence from Additional Experiments:**
> The core idea of LoRA lies in using learnable low-rank matrices to perform incremental updates on the original model, which is similar to our core idea of parameter reusing — using learnable residual matrices (new parameter tokens) to incrementally update the original model. When this idea is specifically applied to parameter efficient tuning, it becomes LoRA. The corresponding specific designs are as follows:
> 1. Freezing the original parameters is intended to save memory costs and ensure parameter efficiency.
> 2. Applying it to linear matrices allows the trained low-rank matrices to be directly merged into the original parameters without adding new inference overhead.
> 3. Partial zero initialization is designed to ensure stable training.
>
> **Implications for Baseline Comparisons:**
> We appreciate the reviewer highlighting this point. In response, we have conducted an additional experiment comparing TokenFormer and Transformer-replacing linear projections with FFNs of equivalent size. The results, shown in below Table, demonstrate that TokenFormer still outperforms this baseline, underscoring the superiority of the Pattention mechanism.
>
> | Method    | model | 124M (60B Scratch) | 354M (15B Reusing)   |
> |:----------------:|:----------------:|:------------------:|:------------:|
> | Net2Net                     | Transformer | 16.4  | 14.3 |
> | Replace Linear with FFN            | Transformer | 16.5  | 14.3 |
> | Ours (Pattention)           | TokenFormer | 16.1 | 13.7 |

---

> ### Author Response · Authors · 2024-12-01
> **Response to Reviewer M5Lm (2/6)**
>
> Below, we address the specific responses one by one.
>
> **A2:**
> We appreciate the reviewer’s agreement with our perspective. It is possible that our models have not yet reached their saturation point, and further investigation is needed to explore the representation span within a fixed vector space in other use cases. As the reviewer noted, extending the saturation point further could be a significant advantage of pattention compared to standard linear projections.
>
> As the reviewer mentioned, the analogy to MoE is exactly because **"MoE models scale the hidden dimension of the base model and then hold this hyperparameter fixed while increasing the number of experts"**. In this context, each key-value parameter pair (or group) in tokenformer could be interpreted as a generalized expert, and TokenFormer scales these experts. This conceptual similarity is why we suggested in our future work that TokenFormer could be extended into an extreme form of MoE, where each key-value parameter pair operates as an individual expert.
>
> Regarding inference costs, we will carefully rephrase the relevant sentences and include additional discussions on short-sequence scenarios of visual modeling (e.g., image classification and CLIP). However, it is worth noting that long-sequence scenarios also have wide applications in visual modeling. For example, high-resolution scenarios such as semantic segmentation. The introduction of Swin Transformer [1] and window-based ViT [2] was precisely to address the enormous computational cost of ViT when processing high-resolution images. There is also an increasing amount of work [3] focused on accelerating vision-language models. For example, FastV [3] addresses the issue of slow inference encountered by LLaVA when handling multiple image inputs or high-resolution vision-language tasks. Long sequence modeling is equally important in vision.
>
> [1] Liu, Ze, et al. "Swin transformer: Hierarchical vision transformer using shifted windows." Proceedings of the IEEE/CVF international conference on computer vision. 2021.
>
> [2] Kirillov, Alexander, et al. "Segment anything." Proceedings of the IEEE/CVF International Conference on Computer Vision. 2023.
>
> [3] Chen, Liang, et al. "An image is worth 1/2 tokens after layer 2: Plug-and-play inference acceleration for large vision-language models." European Conference on Computer Vision. Springer, Cham, 2025.
>
> **A4:**
> A4 was added to address the question regarding the newest baselines as requested by the reviewers, where we included the latest methods (from two months ago) to supplement our experiments. To further address the reviewer's concerns, we additionally conducted experiments replacing the linear layer with FFNs (using an equivalent parameter count to replace the linear layer). The results in the table further validate the effectiveness of our approach.
>
> | Method    | model | 124M (60B Scratch) | 354M (15B Reusing)   |
> |:----------------:|:----------------:|:------------------:|:------------:|
> | Net2Net                     | Transformer | 16.4  | 14.3 |
> | Replace Linear with FFN            | Transformer | 16.5  | 14.3 |
> | Ours (Pattention)           | TokenFormer | 16.1 | 13.7 |
>
> Since we have operations similar to softmax, the scale and variance of the pattention's output is effectively bounded, achieving better results.

---

> ### Author Response · Authors · 2024-12-01
> **Response to Reviewer M5Lm (3/6)**
>
> **A5:**
> Thanks for the reviewer's recognition. Given the laboratory's resources, we are unable to validate TokenFormer at the scale of 300B (e.g., Grok-1). However, it is an academically recognized fact that simply increasing depth is not a sustainable approach for model scaling.
>
> As the reviewer agreed, the additional experiment we provided suggests that this limitation may not apply to TokenFormer. For further clarity, we include another baseline of TokenFormer initialized to the expanded configuration in the below Table. It shows that the scaling results are indeed comparable.
>
> |method | model |hidden dim   | parameter tokens   | layer | param amount | data tokens |final loss |
> |:----------------:|:------------------:|:------------:|:----------:|:----------:|:----------:|:----------:|:----------:|
> |TokenFormer Baseline| TokenFormer | 768        | 1072        | 24  |354M  |15B (scratch)|  2.73  |
> |Transformer Baseline| Transformer | 1024        | -        | 24  |354M       |15B (scratch)|  2.74  |
> |Base model | TokenFormer| 768        | 576        | 12         |  124M | 60B (scratch) | 2.82      |
> |Add param tokens | TokenFormer | 768        | 2140 (+1564)        | 12 (+0)         | 354M | 15B (reusing) | 2.65      |
> |Add param tokens + depth| TokenFormer | 768        | 1072 (+496)        | 24 (+12)  |354M       | 15B (reusing)|  2.64  |
>
> The results in the table show that our extended configuration still performs slightly better than the Transformer under the condition of aligned parameter counts. Additionally, due to parameter reusing, it achieves much better performance than training from scratch when training with the same number of data tokens (test loss: 2.65 vs 2.73), validating the effectiveness of our reusing approach.
>
> **A6:** Since we use a modified softmax (i.e., replacing exponential with gelu), the pattention scores can be negative, making entropy unsuitable as a metric. Instead, our strategy is as follows: for the pattention scores with shape of [batch size, number of parameter tokens], we square the scores and then take the mean along the batch dimension, resulting in a [1, number of parameter tokens] matrix. We then analyze the distribution of the mean corresponding to each parameter token.
>
> | parameter type    | [0,0.5] | [0.5,0.7] | [0.7,0.9]   | [0.9,1.1] | [1.1,1.3] | [1.3,1.5] |[1.5,1.7] | [1.7,1.9] | [1.9,2.1]   | [2.1,2.3] | > 2.3|
> |:----:|:---:|:---:|:---:|:---:|:---:|:---:|:---:|:---:|:---:|:---:|:---:|
> | old param          | 1.5% | 16.2%  | 23.0% | 14.2%|  9.5%| 8.5% | 5.7% |5.9% |3.6%|4.8%   | 7.1% |
> | new param           | 2.3% | 14.2%  | 21.0% | 16.5%|  10.4%| 8.3% | 6.6% |6.1% |5.6%|3.1%   | 5.9% |
>
> The table above shows the distribution of the newly added parameters and the old parameters after finishing the model scaling. It can be observed that the distributions of the new and old parameters are similar, verifying that the newly added parameters did not encounter representation collapse.

---

> ### Author Response · Authors · 2024-12-01
> **Response to Reviewer M5Lm (4/6)**
>
> **A8:** Thanks the reviewer for helping make our explanation clearer.  **We want to clarify that under the premise of the same model parameter count and network depth, $d_0 \cdot n$ will not be greater than $d^2$.** When $d_0 \cdot n > d^2$, it is natural for TokenFormer to have more parameters and therefore a higher computational cost. This is why TokenFormer chooses $n=576$ and $d=768$ to align with the 124M parameter count of Transformer. All our discussions are based on the premise that TokenFormer and Transformer have the same parameter count.
>
> If the network depths of Transformer and TokenFormer are allowed to differ, we acknowledge there is a point where the computational cost of pattention exceeds that of a linear layer. **However, this only occurs when the model size is very small, resulting in an impractical network structure that is extremely shallow and overly wide**. Such configurations are not meaningful, and as the network scale grows, this condition is difficult to satisfy. Additionally, it seems there may be a misunderstanding by the reviewer. The sequence length being processed does not determine whether TokenFormer or Transformer has fewer FLOPs; it only determines how many FLOPs are saved. Once TokenFormer's FLOPs are less than Transformer's, starting from a sequence length of 0, the longer the sequence, the greater the FLOPs savings. Let us clarify.
>
> 1. We analyze this using the two formulas on the right side of the last row in Table 3. If the parameter counts of the Transformer and TokenFormer are the same, meaning $N$ in both formulas is equal, then the additional computational cost of the Transformer compared to the TokenFormer comes only from the token-token interaction, which is $4 \times (n_{\text{layer}}^{\text{transformer}} \times d_{\text{transformer}} - n_{\text{layer}}^{\text{tokenformer}} \times d_{\text{tokenformer}}) \times T^2$. With the model scaling of Transformers, both the depth $n_{\text{layer}}^{\text{transformer}}$ and width $d_{\text{transformer}}$ of the network increase ($n_{\text{layer}}^{\text{transformer}} \times d_{\text{transformer}}$ is larger and larger), whereas TokenFormer expands the model using other dimensions ($n_{\text{layer}}^{\text{tokenformer}} \times d_{\text{tokenformer}}$ is constant), making this advantage increasingly significant. From the above formula, it can also be seen that the sequence length $T$ only determines how many FLOPs are saved; it does not determine whether the computational FLOPs of Pattention are greater or smaller than those of the linear layer.
>
> 2. Indeed, as the reviewer mentioned, there exists a point where $n_{\text{layer}}^{\text{transformer}} \times d_{\text{transformer}}$ < $ n_{\text{layer}}^{\text{tokenformer}} \times d_{\text{tokenformer}}$, but this point typically occurs when the model parameter count is small. Let us clarify. Under the premise of having the same parameter count $N$ of transformer and tokenformer, if we want $n_{\text{layer}}^{\text{transformer}} \times d_{\text{transformer}} < n_{\text{layer}}^{\text{tokenformer}} \times d_{\text{tokenformer}}$ when the model continues to scale up, this requires $n_{\text{layer}}^{\text{transformer}}$ to be as small as possible and $d_{\text{transformer}}$ to be as large as possible. This is because the parameter count of transformer is $12 \times n_{\text{layer}}^{\text{transformer}} \times d_{\text{transformer}}^2$. This model configuration greatly limits the depth of the network, as the reviewer mentioned, which can affect the network's performance. Moreover, as the network's parameter count increases, it will become increasingly difficult to satisfy this condition.
>
> We will include this content in the camera-ready version to make our descriptions more accurate.

---

> ### Author Response · Authors · 2024-12-01
> **Response to Reviewer M5Lm (5/6)**
>
> **A9:** The training improvement (reduced cost) arises from Pattention’s ability to reuse pre-trained parameter tokens, leveraging prior knowledge to achieve the same performance in fewer training steps.
>
> We also provided a method of fixing the hidden dimension and then resuming, as mentioned in A4: replacing the linear layer with FFNs (using an equivalent parameter count to replace the linear layer).
>
> | Method    | model | 124M (60B Scratch) | 354M (15B Reusing)   |
> |:----------------:|:----------------:|:------------------:|:------------:|
> | Net2Net                     | Transformer | 16.4  | 14.3 |
> | Replace Linear with FFN            | Transformer | 16.5  | 14.3 |
> | Ours (Pattention)           | TokenFormer | 16.1 | 13.7 |
>
> Since we have normalization similar to softmax, the scale and variance of the pattention's output is effectively bounded, achieving better results.
>
> Pattention provides a new dimension for scaling, which does not conflict with scaling hidden dimension. In the table below, we provide experiments that scale both the hidden dimension and parameter tokens simultaneously.
>
> | model | strategy | model Param / Training Tokens | hidden dim | param tokens| PPL   |
> |:----------------:|:----------------:|:------------------:|:------------:|:------------:|:------------:|
> |Transformer (base model) |scratch | 124M / 60B | 768 | - | 16.4|
> |Transformer (Net2Net) |reusing | 354M / 15B | 1024 | - | 14.3|
> | TokenFormer (base model) |scratch| 124M / 60B |768| 576  | 16.1  |
> | TokenFormer (add param Tokens) |reusing| 354M / 15B|768| 2140 (+1564)  | 13.7  |
> | TokenFormer (add hidden dim and param tokens) |reusing | 354M / 15B |1024 (+256)| 1536 (+960)  | 13.8  |
>
> From the table above, we can see that our method is compatible with scaling the hidden dimension while also providing an additional scaling dimension beyond the hidden dimension. The advantages compared to previous methods are: (1) faster convergence when incorporating new data (Fig. 6); (2) suitable for long context modeling (Fig. 5); and (3) improved performance (Fig. 7 and above table).
>
> **A10:** As mentioned above, to address the reviewers' concerns, we have removed the analogy to LoRA. However, we want to emphasize that the core idea of LoRA lies in using learnable low-rank matrices to perform incremental updates on the original model, which is similar to our core idea of parameter reusing. Freezing the old parameters is primarily for parameter efficient tuning and memory savings. In my practical use of the successful LLaVA model (for vision-language task with LoRA), not freezing the base model can achieve slightly better performance, but it requires significantly more memory and computational resources.
>
> **A12:** As explained in A8, if TokenFormer and Transformer are required to have the same parameter count and the same number of layers, $n$ can only be smaller than $d$. This is why TokenFormer chooses $n=576$ and $d=768$ to align with the 124M parameter count of Transformer.
>
> Our understanding is that under the condition of Transformers and TokenFormer having the same parameter count, identifying which term is dominant is not critical. Because the $N$ of both are equal and the comparison of FLOPs is only related to $n_{\text{layer}}^{\text{transformer}} \times d_{\text{transformer}}$ and $ n_{\text{layer}}^{\text{tokenformer}} \times d_{\text{tokenformer}}$. Indeed, as the reviewer mentioned, there exists a point that we can adjust depth and width to achieve $n_{\text{layer}}^{\text{transformer}} \times d_{\text{transformer}}$ < $ n_{\text{layer}}^{\text{tokenformer}} \times d_{\text{tokenformer}}$, but this point typically occurs when the model parameter count is small and requires $n_{\text{layer}}^{\text{transformer}}$ to be as small as possible and $d_{\text{transformer}}$ to be as large as possible. Such configurations are not meaningful, and as the network scale grows, this condition is difficult to satisfy. For more explanation, please refer to the response to A8.
>
> Moreover, Figure 5 presents experiments on model scaling starting from a standard 124M base model (depth=12, hidden dim=768) while keeping the network depth unchanged. This setting does not have such a turning point, which makes it meaningful.
>
> We will include this discussion in the camera-ready version.

---

> ### Author Response · Authors · 2024-12-01
> **Response to Reviewer M5Lm**
>
> We sincerely thank the reviewer for the prompt response, allowing us to have a better discussion.
>
> **Comparison with LoRA:**
> We thank the reviewer for pointing this out. From this perspective, using LoRA as an analogy to model scaling is indeed inappropriate. To address this, we have removed the LoRA analogy in the revised version.
>
> We also appreciate the reviewer for highlighting that our A16 results can be analogized to LoRA, as this insight strengthens the contribution of our paper. This analysis, integrated with vision-language discussions, will be included in the camera-ready version.
>
> **A6:**
> We thank the reviewer for the valuable suggestions, which have helped us better formulate this metric.
>
> | parameter type    | [$0$,$4e^{-4}$] | [$4e^{-4}$,$8e^{-4}$] | [$8e^{-4}$,$1.2e^{-3}$]    | [$1.2e^{-3}$,$1.6e^{-3}$]  | [$1.6e^{-3}$,$2.0e^{-3}$] | [$2.0e^{-3}$,$2.4e^{-3}$] |[$2.4e^{-3}$,$2.8e^{-3}$] | > $2.8e^{-3}$|
> |:----:|:---:|:---:|:---:|:---:|:---:|:---:|:---:|:---:|
> | old param          | 14.9% | 39.2%  | 15.5% | 10.6%|  5.7%| 3.8% | 3.5% |6.7% |
>
> | parameter type    | [$0$,$1e^{-4}$] | [$1e^{-4}$,$2e^{-4}$] | [$2e^{-4}$,$3e^{-4}$]  | [$3e^{-4}$,$4e^{-4}$]  | [$4e^{-4}$,$5e^{-4}$]  | [$5e^{-4}$,$6e^{-4}$]  |[$6e^{-4}$,$7e^{-4}$]  | > $7e^{-4}$|
> |:----:|:---:|:---:|:---:|:---:|:---:|:---:|:---:|:---:|
> | new param           | 11.2%| 29.2%  | 20.0% | 13.5%|  8.5%| 6.3% | 4.2% |7.1% |
>
> The table above shows normalized values, where squared scores have been divided by their sum. Since the number of parameter tokens for the new parameters (2140) and the old parameters (576) is different, and the normalized values are around 1/n, we use different scales to depict the two distributions. From the table above, the distributions of the new parameters and the old parameters are roughly similar.
>
> Additionally, we have also followed the reviewer's suggestion to consider the top-1 tokens and compute the normalized entropy over that distribution: diversity = $H / log_2(N)$.
>
> | parameter type    | diversity |
> |:----:|:---:|
> | old param          | 0.34819 |
> | new param          | 0.53536 |
>
> From the table above, it can be seen that the new parameters exhibit even greater diversity, confirming that we did not experience a collapse in representation.
>
>
> **A15:**
> We thank the reviewer for providing this valuable suggestion and perspective. We are currently exploring this direction, and your suggestion has been highly inspiring, greatly aiding the future extension and application of our method.
>
> We sincerely appreciate the time and effort the reviewer has invested in evaluating our work. As we approach the end of the discussion phase, we kindly ask the reviewer to consider if our rebuttal might warrant a reevaluation of the current score. We truly value the reviewer's assessment and are grateful for the constructive feedback provided.

---

> > ### Comment · Reviewer_M5Lm · 2024-12-02
> > **Response to Authors**
> >
> > Thank you for your prompt response and for recomputing the token selection distribution with the diversity measure. I appreciate the additional effort in addressing this aspect of your work.
> >
> > One point of clarification: it was not explicitly stated whether the diversity metric was computed for a representative layer or in aggregate. The former might be more appropriate, as individual selection distributions can shift due to random initialization. To provide greater clarity, I would recommend visualizing the diversity measure as a heat map. This would follow:
> >
> > - **y-axis:** Pattention layers ({Q, K, V, O, F})
> > - **x-axis:** Transformer layers
> > - **z-axis:** Diversity values, fixed between 0 and 1.
> >
> > This would also allow for an intuitive and straightforward visualization of increasing diversity with finetuning, as the colors will become visually brighter, in addition to highlighting any distinct patterns that may emerge. The aggregate modified attention score contributions should still be sufficient, but this additional representation could provide added clarity.
> >
> > From your results, I agree that a strong collapse does not appear to have occurred. While the observed value of 0.35 is somewhat lower than expected, catastrophic collapse typically manifests at diversity levels below 0.1. It is also intriguing to note that adding new tokens improves diversity. This could warrant further investigation, potentially with implications for MoE scaling, though I understand that such exploration is likely beyond the scope of this paper.
> >
> > ---
> >
> > ## Previous Points
> >
> > **A2:** I agree with your comparison, but Tokenformer may involve more than simply "increasing the number of experts." The soft routing behavior of pattention may distinguish it from the hard routing in MoE, which is worth further consideration.
> >
> > Regarding long-sequence image models, I appreciate the additional context. While these are important, your results are based on ViT, a short-sequence model. Extending these findings to long-sequence scenarios is viable but may not scale directly due to differing problem domains.
> >
> > ---
> >
> > **A4:** Thank you for conducting this experiment. The results clearly confirm that Tokenformer’s scaling is not due to the added non-linearity.
> >
> > ---
> >
> > **A5:** I apologize if there was any misunderstanding regarding my intent. I was not asking to confirm scaling with a 300B model. Rather, it was the authors who brought up this example to highlight depth scaling as a challenge. Since the Tokenformer models are at most ~1/3 the depth of Grok-1, this argument seems less relevant in this context. While depth scaling can indeed become unsustainable beyond a certain point, Tokenformer has not yet reached that threshold, nor is there any expectation for the authors to demonstrate an academically recognized behavior.
> >
> > That said, I appreciate the additional baseline experiments with increased depth, both from scratch and with scaling. These results are very promising and contribute significantly to the narrative that Tokenformer achieves better scaling, even with traditional hyperparameters (e.g., not just through the addition of new parameter tokens).
> >
> > ---
> >
> > **A8:** Thank you for clarifying this point. I now understand your argument regarding scaling, where $n<d$ must hold to achieve parameter count parity. To better illustrate this, I encourage revisiting Figure 5 with a focus on measuring the actual FLOPs rather than relying solely on scaling functions. Plotting these in log-log format would make the exponent power and any transition points more apparent, and marking which term dominates in each region of the curves would enhance clarity. This revision would also help substantiate the argument presented in A12.
> >
> > ---
> >
> > **A9:** Thank you for including the additional experiment. It is encouraging to see that Tokenformer continues to exhibit better scaling, even when combined with traditional methods.
> >
> > ---
> >
> > **A13:** Thank you for conducting these evaluations. It is intriguing to see Tokenformer exhibit superior inference time for smaller models, though this advantage seems to diminish and potentially reverse with scale. This aligns with my suspicion that an excessive value of $n$ may create an imbalance across layers. Your results, however, suggest that scaling $n$  less aggressively in favor of traditional parameters such as {$l, d$} could mitigate this issue. While this may be beyond the scope of the current paper, it is a valuable consideration for future work. Moreover, I do not view this as a significant limitation, especially given the recent interest in smaller task-based models where Tokenformer could excel.
> >
> > ---
> >
> > **A17:** Thank you for addressing this point. I agree that the revised phrasing is more appropriate.

---

> > > ### Comment · Reviewer_M5Lm · 2024-12-02
> > > **Final Decision**
> > >
> > > Given the above, I would like to thank the authors for their extensive effort during this discussion. The results have convinced me that Tokenformer exhibits many interesting and desirable properties, and the authors have clarified and isolated the sources of its performance. I believe this work represents a significant contribution to the field.
> > >
> > > While the authors have indicated plans to include a more robust theoretical motivation and revisions to their scaling arguments in the paper, these updates cannot be confirmed or reviewed at this stage. As such, I will raise my score to a 6, reflecting the current state of the paper and results from the discussion. A higher score would have been possible had I been able to evaluate the revised version. I hope the authors can understand my perspective.

---

> > > > ### Author Response · Authors · 2024-12-03
> > > > **Response to Reviewer M5Lm**
> > > >
> > > > We thank the reviewer for the valuable feedback and are pleased to have successfully addressed the concerns raised.
> > > >
> > > > **Diversity Metric:** Thanks for the reviewer’s valuable suggestions. To enhance clarity, we have created a heat map of the diversity measure as requested. The heat map is accessible via this [anonymous link](https://imgur.com/a/RHxUhAe). Please check. From the figure, it can be seen that neither the new parameters nor the old parameters experienced representation collapse, as their diversity values are both greater than 0.1.
> > > >
> > > > **A2:** We thank the reviewer for pointing this out. TokenFormer adds parameters in a "soft" manner, which is indeed different from the hard way used in MoE. We will clarify this point in the final revision.
> > > >
> > > > We thank the reviewer for encouraging us to make our statements more precise. In the camera-ready version, we will more rigorously articulate the advantages of our method in long-context modeling, limiting the scope to the field of autoregressive language modeling.
> > > >
> > > > **A8:** We thank the reviewer’s suggestion. We will make the corresponding modifications and adjustments to Figure 5 in the camera-ready version.
> > > >
> > > > We sincerely thank the reviewer for recognizing the value of our work and for providing extensive and valuable suggestions. Your insightful feedback has played a crucial role in refining our methodologies, clarifying our arguments, and improving the overall presentation of our paper. We deeply appreciate the time and effort you dedicated to reviewing our work.

---

### Official Review · Reviewer_sjss · 2024-11-04

**Soundness:** 3
**Presentation:** 4
**Contribution:** 4
**Rating:** 8
**Confidence:** 4

**Summary:**

This work proposes a new architectural component that can be used in transformers to replace linear projections present in the attention and feed-forward layer. A Pattention layer executes an attention mechanism between the input and some learned key and query tokens to output a transformed token. Both the channel dimensions and the number of key-query tokens can be scaled flexibly, in the sense that smaller models can be used as initial configurations for larger models. Iteratively enlarging the model results in significantly smaller training costs and similar performance to comparable architectures.

**Strengths:**

- Innovative and flexible idea of tokenizing model parameters, that will probably inspire further experimental work in this direction
- Clearly written, well-motivated and contextualized
- Demonstrated lower training cost in comparison of the proposed approach compared to a number of different open-source architectures and datasets

**Weaknesses:**

- The computational complexity of inference and the model itself is only dicussed late in the work.
- The work would benefit from an explicit discussion of the architectures practical limitations.
- See questions below

**Questions:**

- Table 1: "The best performance highlighted for each model" seems to be misleading. It seems it is highlighted in bold for each dataset and model size what the best value is, and even then it is unclear why e.g. LAMBADA acc highlights ours-1.5B and not Pythia 1.8B (this is also the case for several other columns)
- Figure 6: Would you say the (Token)Transformer converged on the test loss respectively (left)?
- It would be nice if the practical relation between parameters, dataset size, scaling stepsize and performance was further investigated. Is there a minimum model size for which this approach only starts being useful? What were your practical considerations in choosing the steps and initializations you did?
- Is there a different effect of scaling the channels and the dimensions?
- Are there artifacts of the scaling steps in the final model? E.g. do the norms of the new channels (new tokens) that were initialized at zero have a different magnitude than those from previous time steps?

---

> ### Author Response · Authors · 2024-11-22
> **Response to Reviewer sjss (1/2)**
>
> We are glad the reviewer found the ideas in the paper innovative, which will probably inspire further experimental work in this direction. Below are our responses to the questions and suggestions raised by the reviewer.
> ***
>
> **Q1: The computational complexity of inference and the model itself is only discussed late in the work.**
>
> **A1:** Thanks for the valuable comments. More discussion has been added to the revision. Please refer to lines 293-295 in the main paper for more details. TokenFormer has two major advantages: controllable token-token computation overhead for efficient long-context modeling and natural suitability for selective token-parameter interaction in sparse inference.
> ***
>
> **Q2: The work would benefit from an explicit discussion of the architectures practical limitations.**
>
> **A2:** The corresponding discussion has been added to the appendix. Please see section G.3 in the appendix. In summary, while Token-Token interaction complexity is controllable, Token-Parameter computation grows linearly with network scale, making it challenging to expand parameters without increasing inference overhead. However, TokenFormer's natural suitability for sparse inference, as each key-value parameter pair functions as an individual expert. This design inspires future integration with MoE to build more efficient and powerful networks.
> ***
>
> **Q3: Table 1: "The best performance highlighted for each model" seems to be misleading. It seems it is highlighted in bold for each dataset and model size what the best value is, and even then it is unclear why e.g. LAMBADA acc highlights ours-1.5B and not Pythia 1.8B.**
>
> **A3:** In Table 1, we present a comparison of the Transformer-based 1.5B and 3B models as a group. To avoid the misleading issue, we have made adjustments in the revised version. Please refer to the updated Table 1 in the main paper.
> ***
>
> **Q4: Figure 6: Would you say the (Token)Transformer converged on the test loss respectively (left)**
>
> **A4:** As explained in Section 4.3, both TokenFormer and Transformer were trained on Enwik8 for a single epoch. TokenFormer demonstrated rapid convergence within this single epoch, while Transformer, which employs the Net2Net method, was unable to effectively resume from the last checkpoint, resulting in large loss jumps and slower convergence. After one epoch, the Transformer had not yet converged, supporting our claim that TokenFormer's scaling method enables faster convergence.
> ***
>
> **Q5: It would be nice if the practical relation between parameters, dataset size, scaling stepsize and performance was further investigated. Is there a minimum model size for which this approach only starts being useful? What were your practical considerations in choosing the steps and initializations you did?**
>
> **A5:** Thanks for this valuable suggestion; this is indeed a direction worth exploring in the future. TokenFormer's scaling method is highly versatile and adaptable to any model size. Currently, we simply follow the standard Transformer configurations: 124M, 354M, 757M, and 1.4B, which are also reported in concurrent works like Mamba.
> ***
>
> **Q6: Is there a different effect of scaling the channels and the dimensions?**
>
> **A6:** Scaling the hidden dimension causes larger loss jumps than the channel (i.e., the number of parameter tokens). The table below shows the loss increases observed at the moment new parameters are added through the hidden dimension and channel respectively. The experiment was conducted in the setting of scaling TokenFormer from 124M to 354M, with the newly added key parameters initialized in two ways: zeros and random initialization.
>
> | scaling type     | original loss    | increased loss (zero init)   | increased loss (random init)   |
> |:----------------:|:----------------:|:-------------:| :-------------:|
> |hidden dim | 2.82             | 5.41 (+2.59)      | 11.77 (+8.95)     |
> |channel   | 2.82             | 2.82 (+0)     | 7.72 (+4.90)      |
>
> As shown in the table above, scaling the channel (increasing parameter tokens) causes smaller loss jumps compared to scaling the hidden dimension. This is because scaling the channel resembles LoRA, learning a residual that minimally impacts the original model's distribution. When key parameters are zero-initialized, like in LoRA, the previously learned distribution is fully preserved, preventing any loss jumps.

---

> ### Author Response · Authors · 2024-11-22
> **Response to Reviewer sjss (2/2)**
>
> **Q7: Are there artifacts of the scaling steps in the final model? E.g. do the norms of the new channels (new tokens) that were initialized at zero have a different magnitude than those from previous time steps?**
>
> **A7:** The table below presents the norm of the new and old key parameter tokens in the last Pattention layer after incrementally scaling the TokenFormer model from 124M to 354M.
>
> | metric | old param tokens | new param tokens |
> |:----------------:|:----------------:|:----------------:|
> |l2 norm| 58.09  | 68.12  |
>
> We observed that even when the key parameters are initially set to zero, they gradually evolve during training, and the new and old parameters eventually converge to the same order of magnitude. This behavior is consistent with experimental observations in LoRA. Moreover, this manner not only fully resumes the previously learned distribution of the smaller model, but also incorporates the additional learning capacity.

---

> > ### Comment · Reviewer_sjss · 2024-11-27
> >
> > I would like to thank the authors for their reply and their answers to my questions. I apprechiate that they were incorporated into the main text and I maintain my score.
> >
> > I do agree with reviewer M5Lm that the main text would profit from a slightly more detailed discussions of the scaling properties. Perhaps the answer to my Q could be extended beyond a setence in a camera ready version of this work.

---

> ### Author Response · Authors · 2024-11-28
> **Official Comment by Authors**
>
> Thank you for your valuable suggestions, which greatly improve the quality of our work. When the new scaling experiments are completed (which we are currently working on, e.g., larger model size), we will include more discussion about the practical relation between parameters, dataset size, scaling stepsize, and performance in the camera-ready version.

---

### Public Comment · ~Yu_Bo1 · 2024-11-14

If I’m understanding correctly, it seems that 'Pattention' is essentially an MLP with some added regularization. It doesn't appear to involve the typical mechanics of an attention mechanism, is that right?

---

> ### Public Comment · ~Xingkui_Zhu1 · 2024-11-20
>
> In my view, the primary mathematical distinction of Pattention compared to FFN lies in its use of alternative activation functions and normalization techniques. However, its most notable innovation is the conceptual shift of treating model parameters as tokens, thereby unifying token-token and token-parameter interactions within a single framework.

---

> ### Author Response · Authors · 2024-11-22
> **Response to Bo Yu**
>
> As mentioned by Xingkui Zhu and the general response, our approach introduces a unified concept for computation by representing all components, including parameters, as tokens. Relationships between these tokens—such as data and parameter tokens—are governed by attention mechanisms.
>
> We start with the formulation that a two-layer MLP can be expressed using attention, and **use this module as the basic computational unit to design the neural network, rather than the linear projection**, thereby unifying all the computations within a single framework.
>
> With this all-attention architecture, TokenFormer enables scalable model expansion without retraining from scratch. Key-value parameter tokens act as learnable patterns, allowing incremental model growth by adding tokens to the parameter set. This method achieves performance comparable to training from scratch but at a lower computational cost.
>
> This tokenization approach extends beyond pretraining and has implications for other areas, such as enabling memory expansion and mitigating information loss in linear models (e.g., Mamba and RWKV)

---

### Public Comment · ~Xingkui_Zhu1 · 2024-11-20
**Comparison Between FFN Zero-Weight Expansion and Tokenformer’s Method**

Tokenformer can be scaled up by adding new key-value token pairs ($K_{\text{new}}, V_{\text{new}}$), allowing incremental expansion without disrupting the pre-trained parameters, which is quite impressive. I was wondering how this approach compares to a more traditional method, such as expanding the FFN hidden dimension by adding zero-initialized weights.

For example, in a standard FFN:
1. The FFN operation is defined as:

   $
   \text{FFN}(X) = \text{Activation}(X W_1 + b_1) W_2 + b_2
   $

   where $W_1 \in \mathbb{R}^{d \times d_{\text{hidden}}}$ and $W_2 \in \mathbb{R}^{d_{\text{hidden}} \times d}$.

2. To expand the hidden dimension $d_{\text{hidden}}$ to $d_{\text{hidden,new}} = d_{\text{hidden}} + \Delta d$, we can augment$W_1$ and $W_2$ as:



   $
    W_1^{\text{new}} =
   \begin{bmatrix}
   W_1 & 0
   \end{bmatrix}, \quad
   W_2^{\text{new}} =
   \begin{bmatrix}
   W_2 \\
   0
   \end{bmatrix}
   $

   Here, the newly added weights are initialized to zeros, and the bias terms $b_1, b_2$ can similarly be expanded with zeros.

3. After expansion, the FFN computation becomes:
4.
   $
   \text{FFN}(X) = \text{Activation}(X W_1^{\text{new}} + b_1^{\text{new}}) W_2^{\text{new}} + b_2^{\text{new}}
   $

   Substituting the expanded matrices:

   $
   \text{FFN}(X) = \text{Activation}(X W_1 + b_1) W_2 + b_2
   $

   This shows that the output remains identical to the original computation because the newly added weights contribute nothing initially.

5. During subsequent training, the zero-initialized weights are gradually optimized to contribute, effectively expanding the model’s capacity.

---

Could you please elaborate on the differences between this FFN zero-expansion approach and your method of adding key-value tokens? Specifically:
- How do the two approaches differ in their flexibility for model scaling?
- From a computational efficiency perspective, does your method have advantages in terms of memory or training speed?

Thank you for your insights!

---

> ### Author Response · Authors · 2024-11-22
> **Response to Xingkui Zhu**
>
> Thank you for raising this very valuable question. First, if both layers are zero-initialized, it would lead to a decrease in expressive capability. To ensure diverse gradients, at least one layer must be randomly initialized, or alternatively, all layers are randomly initialized.
>
> The table below presents a performance (i.e., perplexity) comparison of FFN expansion. To maintain fairness with Net2Net (random initialization), all new parameters in these methods were randomly initialized. The FFN expansion method underperforms compared to our approach and even slightly falls short of Net2Net.
>
> | Method    | model | 124M (60B) | 354M (15B)   |
> |:----------------:|:----------------:|:------------------:|:------------:|
> | Net2Net        | Transformer | 16.4 | 14.3 |
> | FFN Expansion | Transformer | 16.4  | 14.5 |
> | Ours           | TokenFormer | 16.1 | 13.8 |
>
> The table shows that expanding only the FFN slightly degrades performance, likely due to the lack of scaling in the qkv and output projection layers. This imbalance limits the expressive power of the self-attention block and weakens token-token attention. We argue that parameter scaling should be uniform—each token-parameter interaction module should scale proportionally. Otherwise, unscaled components create bottlenecks in the model's expressive capability.
>
> The FFN expansion approach is a simplified version of our method, where only the FFN is replaced with Pattention, while the other linear projection components remain unchanged.
> Therefore, in terms of model flexibility, it is inferior to TokenFormer. From a computational efficiency perspective, the training time of a Transformer, the FFN expansion approach, and TokenFormer are similar, being roughly proportional to the number of parameters.

---

> ### Public Comment · ~Xingkui_Zhu1 · 2024-11-25
>
> I sincerely appreciate the authors' thorough response and the considerable effort they put into addressing my questions with additional analysis.

---

### Author Response · Authors · 2024-11-22
**Response to All Reviewers**

We thank all reviewers for their careful reading of our work and their thoughtful feedback. All reviewers acknowledged that our work makes a significant architectural contribution, is clearly written, well-motivated, and demonstrates strong empirical results.

Our paper provides a novel perspective for computation where all components, including parameters, are represented as tokens. Interactions between different token types are handled by the attention mechanism. Then, all the linear projections in the Transformer are replaced with this design. This innovation allows for greater architectural flexibility and supports incremental model scaling without retraining, substantially reducing training costs.

Reviewers posed insightful questions and constructive comments, which we have addressed in detail in our individual responses.

***

**Discussion of the two-layer MLPs and our pattention.**

**A:** Our motivation is to provide a novel and unified perspective to model computation where all the components, including parameters, are represented as tokens. Interactions between different token types, such as data tokens and parameter tokens, are managed using the attention mechanism.

Specifically, we start with the formulation that a two-layer MLPs can be implemented as attention [1], and **use this module as the basic computational unit to design the neural network, rather than the previous linear projection**, thereby unifying token-token and token-parameter interactions within a single framework.

Based on this all-attention property, TokenFormer naturally supports incremental scalability, enabling seamless expansion without retraining from scratch. We treat a pair of key-value parameter tokens as a learnable pattern and incrementally scale the model by augmenting this parameter set. This approach enables our model to progressively grow into a larger model based on a pre-trained smaller model, achieving performance comparable to training from scratch but at a significantly lower training cost.

This concept of tokenizing everything and using attention to establish relationships has the potential to impact topics beyond pretraining models. For example, it can be applied to the hidden states of linear models, enabling incremental memory expansion and reducing excessive information loss observed in linear models (e.g., Mamba [2] and RWKV [3]).


[1] Geva, Mor, et al. "Transformer feed-forward layers are key-value memories." EMNLP 2021.

[2] Gu, Albert, and Tri Dao. "Mamba: Linear-time sequence modeling with selective state spaces." COLM 2024.

[3] Peng, Bo, et al. "Rwkv: Reinventing rnns for the transformer era." EMNLP 2023.

---

### Meta-Review · Area_Chair_EaeA · 2024-12-19

**Metareview:**

Summary
This paper introduces TokenFormer, an innovative framework that replaces linear layers in transformers with a novel "Pattention" mechanism. The proposed architecture tokenizes model parameters and employs attention to facilitate interactions between input tokens and model parameters, enabling incremental scaling of models without retraining from scratch. This approach significantly reduces training costs while maintaining or surpassing baseline performance. The paper is well-executed and presents strong empirical results, making a substantial architectural contribution to model scaling and efficient resource utilization.

Decision
Based on the reviews and discussions, I recommend accepting this paper for a spotlight presentation. The paper is a significant step forward in scalable and efficient transformer architectures, and its findings are likely to inspire further research.

**Additional Comments On Reviewer Discussion:**

Reviews and Discussion
All reviewers acknowledged the paper's innovation, clear writing, and robust experimental validation. Key strengths include the novel concept of tokenizing parameters, efficient scaling, and reduced training costs. Reviewers also appreciated the thoughtful responses from authors, which clarified initial ambiguities in the mathematical formulation and addressed additional baselines like HyperCloning. Some concerns were raised about limited theoretical backing, scalability to larger models, and comparison methodologies, but the authors provided comprehensive rebuttals and additional experiments, which strengthened the paper's position.

---

### Decision · Program_Chairs · 2025-01-22

Accept (Spotlight)